# Harnessing Optimization Dynamics for Curvature-Informed Model Merging

## Abstract

Model merging is an effective strategy for composing capabilities in large language models without the need for costly joint retraining. We study this process in the supervised fine-tuning (SFT) stage, consolidating multiple checkpoints specialized for distinct capabilities (e.g., math, coding, and precise instruction following) into a single model. First, we introduce Optimization Trajectory Aware (OTA) Merging, a curvature-aware method for mitigating task interference that uses optimizer second-moment statistics as a diagonal curvature proxy to first prune the task vector with our Fast Fisher Grafting (FFG) technique and then reweight the pruned vector. When merging diverse, capability-based checkpoints, OTA improves the merged model's performance over strong baseline methods, as evaluated on unseen capability-based benchmarks. Second, we conduct a comprehensive, theoretically-inspired empirical analysis to explain the effectiveness of OTA. Our analysis surprisingly reveals that FFG implicitly induces a layer- and role-wise aware pruning mechanism that is capable of maintaining fine-tuning performance at much more aggressive pruning ratios compared to magnitude pruning and that exhibits interpretable task localization properties. Third, an extensive comparison of our curvature proxy across capability checkpoints shows that experts converge to a basin with substantial curvature similarity, offering a novel lens on why simple linear merging can be effective in practice. This result further strengthens our ablation study, showing that FFG is critical for merging performance. Finally, we develop a memory-light variant of OTA that efficiently compresses the second moments, mitigating the additional storage requirements of our method and improving scalability. We make all code, training and evaluation scripts, visualization artifacts, and capability-specific SFT checkpoints accessible through an anonymized repository at `https://github.com/tmlr-ota/ota`.

## 1 Introduction

Large language models (LLMs) have achieved remarkable success as generalist foundations for diverse tasks, with fine-tuning on specialized data yielding expert models that excel in targeted domains Brown et al. (2020). However, deploying an ever-growing suite of specialized experts incurs prohibitive operational and computational costs, motivating research into model merging—consolidating multiple expert capabilities into a single multitask model without retraining costs or ensembling latency.

Despite empirical successes ranging from weight averaging to curvature-aware methods like Fisher Merging (Matena & Raffel, 2022b), the fundamental question remains: why does model merging work? The prevailing hypothesis—that fine-tuned models co-inhabit a single, wide, flat loss basin enabling linear model connectivity (Frankle et al., 2020)—fails at non-trivial scales and for models trained on disparate tasks with distinct optimization trajectories. Yet simple linear averaging remains competitive against sophisticated methods at scale (Yadav et al., 2024), revealing a critical gap in our understanding of loss landscape curvature and limiting theoretical guidance for merging strategies.

We present a novel empirically-grounded perspective on SFT fine-tuned LLM curvature. Our central insight: second-moment estimates (`exp_avg_sq`) from adaptive optimizers like Adam Kingma & Ba (2014) serve as powerful, readily available proxies for the Fisher information matrix diagonal and loss landscape curvature. We

operationalize this through the **Optimization Trajectory Aware (OTA)** merging framework, employing a two-stage process. First, **Fast Fisher Grafting (FFG)** leverages optimizer states to identify and revert noisy parameter updates, restoring non-essential changes to base model values—a principled grafting approach Panigrahi et al. (2023). Figure 1 (left) reveals that task-specific knowledge exhibits high localization and structured sparsity. Second, OTA aggregates denoised experts via curvature-aware merging using the same optimizer states as preconditioners. Our results (Figure 1, right) demonstrate consistent outperformance of established baselines across diverse capabilities, with FFG's saliency-aware denoising driving the most significant gains. To mitigate heavy data curation stages for post-training and ensure experimental reliability, we fully replicated the SFT stage of Tulu3 Lambert et al. (2024), an open-source post-training pipeline built on Llama 3 models.

Our extensive experiments reveal FFG's distinct localization patterns compared to magnitude pruning: aggressive **structured column/row sparsity** in early and late query/key layers, and specialized attention heads in late-layer output projections. To address storage requirements for second-moment matrices (comparable to model size), we propose rank-one **AdaFactor-style** compression Shazeer & Stern (2018), demonstrating maintained performance on merging benchmarks. Low stable rank across transformer layers validates this compression. Finally, we provide **compelling empirical evidence for a new merging theory**: capability-based SFT checkpoints develop shared curvature geometry explaining linear averaging success, with models trained on identical data but different learning rate schedules exhibiting **nearly identical curvature structures**.

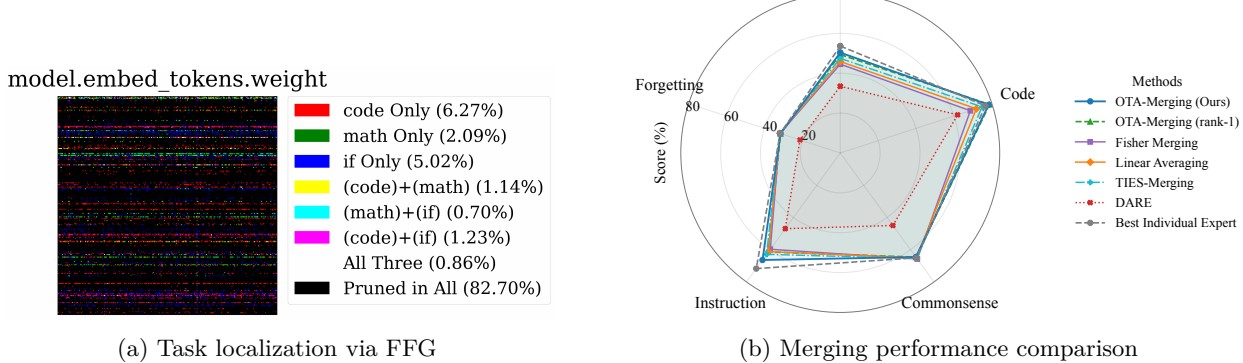

(a) Task localization via FFG

(b) Merging performance comparison

Figure 1: **Left:** FFG reveals that task-specific knowledge is highly localized. This heatmap shows the FGG mask regions for three expert models (math, code, instructions) in the token embedding layer, demonstrating clear, low-rank structured sparsity induced by FFG at 40% global density. **Right:** A capability-based comparison shows that our full OTA method, which combines FFG-based denoising with curvature-aware aggregation, is the top-performing merging technique. The dashed line represents the performance ceiling of the best individual expert for each capability.

## 2 Related Work

**Weight-Space Model Merging and Composition.** A rapidly growing literature studies how to combine separately fine-tuned models directly in weight space. Early work showed that simple weight averaging often improves accuracy and robustness when fine-tuned solutions lie in a shared basin (Wortsman et al., 2022). Matena & Raffel (2022a) formalize merging as approximate posterior combination via Fisher-weighted averaging, where (diagonal) Fisher information acts as parameter-wise preconditioner. Task arithmetic composes behaviors by adding/subtracting task vectors (Ilharco et al., 2022); its theory and practice were strengthened by Ortiz-Jimenez et al. (2023), who advocate editing in the model's tangent space. Permutation alignment methods such as Git Re-Basin expose linear connectivity by matching hidden units before interpolation (Ainsworth et al., 2023). To curb interference, TIES-Merging trims small edits and resolves sign conflicts (Yadav et al., 2023). More recently, Tam et al. (2024) cast merging as solving a linear system in a task-parameter subspace (MaTS), while Huang et al. (2024) propose a tuning-free, high-performance recipe

(EMR) that works across modalities. Practitioner tooling such as MERGEKIT has standardized many of these strategies for LLMs (Goddard et al., 2024b). Our approach complements these directions with a curvature-aware, two-stage pipeline: (i) FFG selects/denoises per-parameter edits using the second moments combined with Optimal Brain Surgeon methodology Hassibi & Stork (1992a) , and (ii) curvature-preconditioned aggregation reweights surviving edits during merging.

**Curvature Proxies from Optimization Dynamics.** Our work repurposes the readily available second-moment statistics from adaptive optimizers as a proxy for the diagonal Fisher information. Recent work by Li et al. (2025) compellingly validates this core idea, introducing the "Squisher" and demonstrating its effectiveness as a "for free" replacement for a calculated Fisher across a broad set of applications, including model pruning, continual learning, and a form of Fisher-merging. While our work shares this foundational insight, it diverges significantly in its methodology, application focus, and conceptual contributions. We detailed this, alongside additional related works, in Appendix A.

## 3 The OTA-Merging Framework

We propose OTA Merging, a unified framework designed to merge fine-tuned experts by addressing parameter interference and curvature misalignment in a principled, storage-efficient manner. Our approach is built on a key insight: the second-moment estimates tracked by adaptive optimizers like Adam Kingma & Ba (2014) can serve as a computationally cheap yet effective proxy for the local curvature of the loss landscape. By leveraging this curvature information, OTA-Merging executes a three-stage process: (1) it identifies and isolates the critical parameters for each task using a novel pruning strategy, FFG; (2) it aggregates these task-specific subnetworks using a curvature-aware weighting scheme; and (3) it employs a compression technique to store the required second moment information with minimal memory overhead.

**Adam's Second Moment as a Proxy for the Empirical Fisher.** Preconditioning-based optimizers, such as Adam Kingma & Ba (2014) and AdaGrad Duchi et al. (2011), scale gradients by a preconditioner matrix that approximates the Fisher Information Matrix (FIM). For a model with parameters $\mathbf{w}$, the update at step $k$ is given by $\mathbf{w}_{k+1} = \mathbf{w}_k - \eta \mathbf{P}_k^{-1} \mathbf{m}_k$, where $\eta$ is the learning rate, $\mathbf{m}_k$ is the first momentum of the gradients, and $\mathbf{P}_k = \text{Diag}(\mathbf{v}_k)$ is a diagonal preconditioner derived from the second moment, $\mathbf{v}_k$.

The second moment, $\mathbf{v}_k$, is typically an exponential moving average (EMA) of element-wise squared gradients: $\mathbf{v}_k = \beta_2 \mathbf{v}_{k-1} + (1 - \beta_2)(\nabla \mathcal{L}_{B_k}(\mathbf{w}_k))^{\odot 2}$, where $B_k$ is the mini-batch at step $k$. This formulation means that $\mathbf{P}_k$ accumulates information about the diagonal of the empirical FIM over the optimization trajectory. A comprehensive study on the connection between the empirical FIM and the Hessian is provided by Martens (2020). Moreover, Morwani et al. (2024) studies the connection of the outer product of mini-batch gradients to the empirical FIM, and Li et al. (2025) further validates the effectiveness of second moments as a Fisher proxy. By leveraging these works, the connection between the second moment and the Hessian can be formalized with detailed theoretical justification, and proofs are deferred to Appendix C for completeness.

**Component 1: Parameter Selection with FFG.** To mitigate destructive interference when merging, we first identify a subnetwork within each expert that maintains the fine-tuning performance of the full model. Inspired by Optimal Brain Damage LeCun et al. (1989), we score the saliency of each parameter's change from its pre-trained state $\mathbf{w}_0$. The saliency of a parameter change $\Delta w_{\tau,i} = w_{\tau,i}^* - w_{0,i}$ for task $\tau$ is defined by its contribution to the loss, approximated by a second-order Taylor expansion: $s_{\tau,i} = \frac{1}{2}\mathbf{H}_{ii}(\Delta w_{\tau,i})^2$.

Calculating the Hessian $\mathbf{H}$ is infeasible for large models. However, the second-moment estimate $\mathbf{v}_\tau$ from the Adam optimizer serves as an effective, training-free proxy for the diagonal of the FIM, which in turn approximates the Hessian. This insight leads to our FFG saliency score, defined as $s_{\tau,i} = (\Delta w_{\tau,i})^2 \cdot v_{\tau,i}$. For each expert $\tau$, we compute this score for every parameter in its task vector $\Delta \mathbf{w}_\tau = \mathbf{w}_\tau^* - \mathbf{w}_0$. We then generate a binary mask $\mathbf{m}_\tau$ by preserving only the top-$k$ parameters with the highest saliency scores, where $k$ is set by a sparsity ratio $\rho$. Instead of pruning parameters to zero, we graft by reverting the non-selected parameters back to their $\mathbf{w}_0$ values. The resulting pruned task vector is thus $\Delta \mathbf{w}_\tau' = \mathbf{m}_\tau \circ \Delta \mathbf{w}_\tau$.

**Component 2: Curvature-Aware Aggregation.** After identifying the essential subnetwork for each expert, we must aggregate them in a manner that respects the curvature of the loss landscape. Inspired by

preconditioned model merging methods such as Fisher Merging Matena & Raffel (2022a) (see Appendix B for additional background), we achieve this by solving for a merged parameter vector that is at minimal distance from each of the pruned task vectors, where distance is measured in a space warped by the curvature. Let $\mathbf{P}^*_{\tau,\mathrm{Adam}} = \mathrm{Diag}(\sqrt{\mathbf{v}^*_\tau} + \epsilon)$ be the diagonal preconditioning matrix derived from Adam's second-moment estimates for expert $\tau$. The merged model is the solution to the following optimization problem: $\mathbf{w}_{\mathrm{merged}} = \mathbf{w}_0 + \arg\min_{\Delta\mathbf{w}} \sum_{\tau=1}^{T} \|\Delta\mathbf{w} - \Delta\mathbf{w}'_\tau\|^2_{\mathbf{P}^*_{\tau,\mathrm{Adam}}}$. This objective has a closed-form solution, yielding a pre-conditioned average of the pruned task vectors:

$$\mathbf{w}^{\mathrm{OTA}}_{\mathrm{merged}} = \mathbf{w}_0 + \left(\sum_{\tau=1}^{T} \mathbf{P}^*_{\tau,\mathrm{Adam}}\right)^{-1} \left(\sum_{\tau=1}^{T} \mathbf{P}^*_{\tau,\mathrm{Adam}}(\mathbf{m}_\tau \circ \Delta\mathbf{w}'_\tau)\right). \tag{1}$$

This unified equation elegantly demonstrates how OTA first determines *what* to merge via the FFG mask $\mathbf{m}_\tau$ and then decides *how* to merge using the compressed, curvature-aware preconditioner $\hat{\mathbf{P}}^*_\tau$, forming a complete and scalable framework.

**Component 3: Memory-Efficient Preconditioner Compression.** A practical challenge is that storing the full second-moment tensor $\mathbf{v}_\tau$ for each expert doubles the storage cost. To overcome this, we adopt a compression strategy inspired by AdaFactor Shazeer & Stern (2018). For any large weight matrix, instead of storing the full $\mathbf{v}_\tau$, we only store the moving averages of its row-wise and column-wise sums. We can then reconstruct a non-negative, rank-1 approximation of the second-moment tensor, $\hat{\mathbf{v}}_\tau$, from these compressed statistics at runtime. This low-rank approximation is then used to form a compressed preconditioner, $\hat{\mathbf{P}}^*_\tau = \mathrm{Diag}(\sqrt{\hat{\mathbf{v}}_\tau} + \epsilon)$, which replaces its full-rank counterpart in both the FFG saliency calculation (Section 3) and the OTA aggregation formula (Eq. 1). For additional background on AdaFactor, see Appendix B. Moreover, we would like to highlight that approaches such as SVD would not be effective here, as we are factorizing a non-negative matrix.

# 4 Experiments

We evaluate OTA-FFG across diverse benchmarks, studying how FFG localizes task-critical parameters and how OTA aggregates them. We compare against strong baselines and magnitude pruning across sparsity levels, and analyze mask structure and curvature to explain observed gains and compression benefits. Moreover, we analyze second-moment curvature of SFT models, finding highly similar curvature across capabilities and near-identical curvature under different schedulers, motivating why simple linear averaging works.

## 4.1 Experimental Setup

**Models, Tasks, and Training.** Our experiments use `meta-llama/Meta-Llama-3.1-8B` as the base model. To create a realistic merging scenario, we fine-tune five SFT models on distinct, capability-aligned subsets of the `allenai/tulu-3-sft-mixture` dataset Lambert et al. (2024). These capabilities include mathematics (using `Tülu 3 Persona MATH`, `OpenMathInstruct 2`, and `NuminaMath-TIR`), coding (using `Tülu 3 Persona Python` and `Evol CodeAlpaca`), general instruction following (using `WildChat (GPT-4 subset)`, `OpenAssistant`, and `No Robots`), knowledge recall (using `FLAN v2`, `SciRIFF`, and `TableGPT`), and precise instruction following (using `Tülu 3 Persona IF`). This setup creates a well-posed aggregation problem where each expert localizes a complementary skill.

All models are fine-tuned using full-parameter SFT via the `LLaMA-Factory` library Zheng et al. (2024). Crucially for our method, we use the AdamW optimizer and save the complete optimizer state, including the exponential moving average of squared gradients (`exp_avg_sq`), which serves as our preconditioning tensor and curvature proxy.

**Methods Under Comparison.** We evaluate our proposed method and its ablations against a suite of strong baselines implemented in `MergeKit` Goddard et al. (2024a). We evaluate OTA-FFG (ours), OTA without FFG, FFG-TA (FFG + linear averaging), and baselines—Linear, TIES, DARE, Breadcrumbs, Fisher—using MergeKit implementations.

**Evaluation Suite.** We evaluate all merged models on a diverse set of benchmarks using the Tülu-3 evaluation suite via the OLMES toolkit Lambert et al. (2024), ensuring a rigorous and reproducible assessment. The suite includes: **HumanEval(+)** Chen et al. (2021); Liu et al. (2024) for coding, **GSM8K** Cobbe et al. (2021) and **MATH** Hendrycks et al. (2021) for mathematical reasoning, **IFEval** Zhou et al. (2023) for instruction following, **BBH (CoT)** Suzgun et al. (2022) for general reasoning, **DROP** Dua et al. (2019) for reading comprehension, and **PopQA** Mallen et al. (2023) for knowledge recall. Moreover, to further strengthen the reliability of our experimental setup, each capability-based SFT checkpoint is evaluated on the entire evaluation suite, and as expected we observe that each SFT model indeed excels at the unseen evaluation benchmark corresponding to its capability. The evaluation results are provided in Table 4 in Appendix D.4.

## 4.2 Main Results: Merging Performance

**OTA with FFG Achieves State-of-the-Art Merging Performance.** The main results in Table 1 confirm our core hypothesis. We compared the performance of several methods for merging the five SFT checkpoints discussed previously. The table shows the performance of each merging method (rows) on a specific capability (columns). A capability's performance is measured by averaging the scores from the benchmarks assigned to it in our evaluation suite. Specifically: Math performance is the average of the GSM8K and MATH benchmarks; Code is the average of HumanEval and HumanEval+; Commonsense is the average of BBH and Drop; Instruction-Following is measured by IFEval; and Forgetting is measured by PopQA. Our full method, OTA, achieves the highest average score (0.582) across all merging techniques, outperforming strong baselines like TIES (0.565). The ablation studies clearly show that the most significant gains come from FFG's saliency-based task vector sparsification. The FFG-TA (Selection Only) ablation, which simply averages FFG-pruned task vectors, already achieves a strong 0.560 average. This is substantially better than OTA (Aggregation Only) (0.536), which uses our curvature preconditioning on unpruned task vectors. This result strongly supports our thesis that the primary obstacle in merging non-IID experts is parameter interference, which FFG effectively mitigates by acting as a denoiser. The poor performance of DARE (0.417) further reinforces that naive, random pruning is detrimental; a saliency-aware method is essential. Moreover, for all methods under comparison, the sparsity ratio is tuned on a per-expert basis, whether using FFG, magnitude or random pruning on task vectors. We observed that tuning a fixed sparsity ratio for all experts made the performance of both OTA and TIES no better than that of linear merging. Moreover, we evaluated the Tulu3-SFT, which can be seen as a multi-task SFT model trained on aggregated data of all capability-based SFT checkpoints. We observed that there is still a considerable but not significant gap between capability-based SFT merging and multi-task SFT training, strengthening the effectiveness of model merging in post-training and showing room for further improvement of merging methods.

Table 1: Performance comparison of merging methods. The best-performing merge method in each column is highlighted. The "Average" score is the unweighted mean across the five capability metrics.

| Model | Math | Code | Commonsense | Instruction | Forgetting | Average |
|---|---|---|---|---|---|---|
| DARE | 0.335 | 0.619 | 0.450 | 0.470 | 0.212 | 0.417 |
| Breadcrumbs | 0.453 | 0.722 | 0.547 | 0.529 | 0.260 | 0.502 |
| Fisher | 0.446 | 0.686 | 0.657 | 0.597 | 0.318 | 0.541 |
| Linear | 0.459 | 0.718 | 0.650 | 0.612 | 0.318 | 0.551 |
| TIES | 0.475 | 0.748 | **0.654** | 0.629 | **0.318** | 0.565 |
| OTA (w Linear) | 0.458 | 0.771 | 0.650 | 0.601 | 0.318 | 0.560 |
| OTA (wo FFG) | 0.458 | 0.660 | 0.654 | 0.590 | 0.318 | 0.536 |
| OTA (rank1) | 0.494 | **0.787** | 0.646 | 0.614 | 0.315 | 0.571 |
| OTA | **0.504** | 0.783 | 0.645 | **0.664** | 0.315 | **0.582** |
| Tulu3-SFT | 0.528 | 0.835 | 0.650 | 0.715 | 0.295 | 0.605 |

## 4.3 Deep Dive: Analysis of the FFG Stage

**FFG Consistently Outperforms Magnitude Pruning.** To validate the FFG mask selection mechanism, we compare it directly against magnitude pruning across a range of density ratios (see Figure 7 in Appendix E).

We apply FFG and magnitude pruning on task vectors of math, code, and precise-if SFT models, and evaluate each expert on its corresponding benchmark. FFG consistently matches or outperforms magnitude pruning, with the largest gains observed in high-sparsity regimes (1–10% density). For instance, on IFEval, FFG yields a +0.10 to +0.16 absolute improvement at 1–5% density. On the Code benchmark (HumanEval), FFG at 20% density (0.834) even surpasses the dense SFT model (0.788), suggesting that FFG has a regularizing effect by removing noisy, low-saliency updates and thereby improving generalization. A similar pattern is observed for the math SFT model on the MATH benchmark, where at 40% density FFG achieves 32.52%, compared to the full math SFT performance of 31.6%. The ability of FFG to compress task vectors to much higher sparsity levels while still maintaining, or even improving, fine-tuning performance further motivates an analysis of its underlying subnetwork selection mechanism. Hence, in the subsequent section, we provide a comprehensive empirical study to better understand this mechanism.

### 4.3.1 Analyzing the Underlying Mechanism of FFG

To investigate the distinct mechanisms of magnitude pruning and FFG, we applied a global density ratio of 40% to generate sub-network masks for SFT models trained on math, code, and precise-if tasks—a budget at which both methods maintain fine-tuning performance. We analyzed the resulting layer-wise and role-wise density patterns (across weight type, and layer depth) induced by global density.

Figure 2 plots the density distributions for the math SFT model, revealing a U-shaped sparsity pattern across transformer layers for both methods, with aggressive pruning in early and late layers while middle layers remain dense. Query and key projections share similar sparsity patterns within each method, as do up- and down-projection weights. FFG exhibits dramatically wider sparsity ranges than magnitude pruning: in the math SFT model, FFG sparsity spans 99% (first layer's query projection) to 18% (fifth layer's value projection), while magnitude pruning ranges from 72% (first layer's value projection) to 45% (layer 15 query weight).

The methods exhibit opposing density rankings across weight types. Magnitude pruning preserves query and key projections (lower sparsity) while aggressively pruning value and output projections (higher sparsity). **Surprisingly**, FFG inverts this pattern: it aggressively prunes query and key weights—particularly in initial and late layers (99% sparsity for first-layer query/key weights versus 70% for magnitude pruning)—while densifying value and output projections in middle layers (18% sparsity for layer 5 value projection versus 58% for magnitude pruning). This same reversal occurs for down- and up-projections, with FFG showing more extreme U-shaped patterns.

The aggressive query/key sparsification achieved by FFG motivated further structural analysis. Computing row-wise and column-wise densities for the mathematical SFT model revealed that FFG induces highly structured sparsity: over 85% of query matrix rows (output features) in the four layers exhibiting the highest sparsity are entirely zeroed (see Figure 12 in Appendix E). Magnitude pruning lacks this structure (see Figure 13 in Appendix E), suggesting that **low-rank sparsity** is an implicit property of FFG.

**FFG shows an implicit layer-wise and role-wise grafting mechanism** Overall, our analysis of the density distribution patterns reveals novel insights into the task localization of FFG across weight types and layer depths and strongly supports its implicit layer-wise density allocation. FFG aggressively sparsifies the early and late query/key layers, even at a moderate global density ratio (e.g., 40%), and imposes a low-rank structure on their task vectors. On the other hand, it allocates most of the global density to the value and projection weight types in the middle layers (approximately twice the density allocated to these weight types compared to magnitude pruning). This implicit density allocation mechanism aligns well with our understanding of SFT training paradigms, where the query and key layers of task vectors were shown to be extremely low-rank, as presented in the seminal work LoRA (Hu et al., 2022) and further studied theoretically in Tarzanagh et al. (2023).

### 4.4 SFT Task Localization through FFG lens

Inspired by FFG's layer-wise and role-wise grafting, we analyze SFT task localization by comparing FFG masks/sub-networks across math, code, and precise-if models within each layer and weight type (Figure 3).

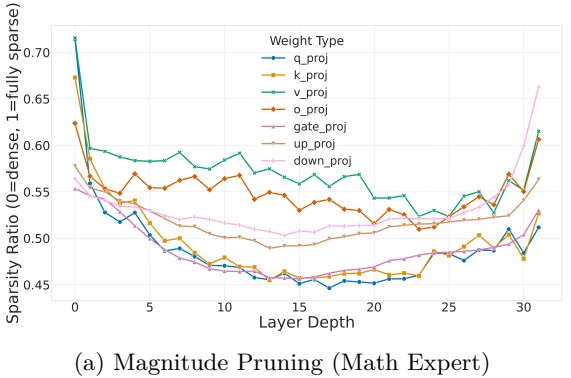
(a) Magnitude Pruning (Math Expert)

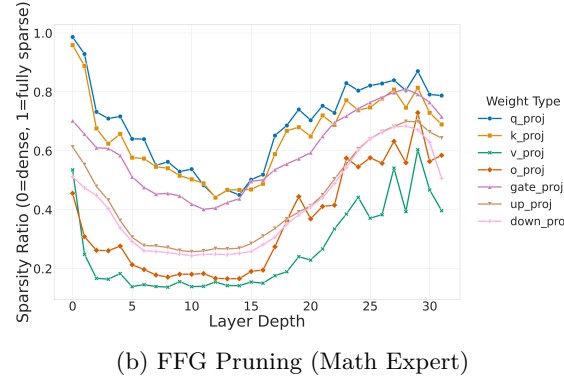
(b) FFG Pruning (Math Expert)

Figure 2: **Layer-wise density distribution at a global 40% task vector pruning density for the Math SFT expert.** FFG (right) exhibits strong, emergent role-aware pruning, aggressively sparsifying query/key weights while preserving value/output/FFN weights. Magnitude pruning (left) is far more uniform and less structured.

With FFG density fixed at 40% across experts, visualizations show mask localization where colors indicate element status: dark for pruned from all three tasks, white for selected by all, and other colors for partial overlap. Weights are downsampled (e.g., $4096 \times 4096$ to $256 \times 256$) via uniform subsampling with adaptive stride. We visualize layers 1, 15, and 30 as representative examples; however, we discuss only patterns that are consistent across all layer depths within each weight type. For completeness and to establish the credibility of our claims, we provide heatmap visualizations across all layer depths and weight types, which are accessible through the provided anonymous GitHub repository.

Figure 3 legends show computed mask overlap between SFT models. In all heatmaps (excluding embedding and LM heads), columns represent single input feature weights to all outputs, while rows represent all weights to single outputs. Thus, dense/sparse colored rows indicate output features densely/sparsely utilizing inputs, while dense colored columns show heavily updated input features during SFT and dark columns represent unused inputs.

**SFT Use a Shared, Extremely Sparse Subset of Embeddings' Features Across Tasks.** As shown in Section 4.3, FFG introduces the most sparsity in the first two and last two layers of the transformer. The mask visualizations for these layers are predominantly dark, with strong row- and column-wise patterns, which supports the low-rank structure of the FFG mask in these areas. Most interestingly, the query and key weights in the first layer show extremely high overlap in pruned regions across the three tasks (e.g., a 97.8% shared pruned region). We observed that all three masks select an extremely sparse and nearly identical set of input features (a column-wise mask pattern) for the query and key matrices of the first layer, with almost no dense output features. This strongly suggests that SFT updates only a very sparse subset of embedding features. This observation is reasonable, as we expect the early layers to be focused on general language understanding, an ability largely acquired during pre-training.

**Task-Specific Dominance in Attention vs. FFN Layers.** The Math SFT model dominates query and key attention layers across all depths with often twice the parameter density of Precise-IF SFT. Code SFT similarly exceeds Precise-IF in these layers, though less dramatically. This pattern peaks in middle layers. Conversely, Precise-IF and Code SFT dominate FFN layers (up, down, gate projections)—clearly in layers 1-2, absent until layer 22, then aggressively from layers 23-31. In these later layers, Precise-IF significantly exceeds both Code and Math SFTs' density, with Code slightly denser than Math. Value and output projections follow the ranking Math > Code > Precise-IF (slight differences). This aligns with expectations: attention layers are critical for mathematical reasoning's complex token patterns, while instruction-following relies more on FFN layers' feature extraction capabilities.

**Formation of Specialized and General-Purpose Attention Heads in Layers 17-31.** The value and output projections reveal another interesting property. From layers 1 to 16, we observed maximum dense

mask overlap across the expert models compared to other weight types, and the heatmaps appear mostly random with no clear visual pattern. However, from layers 17 to 31, two distinct regions emerge within the masks. The first region maintains high overlap across the experts, while the second region contains almost no overlap. This phenomenon is illustrated for the value and output projections of layer 30 in Figure 3. It is worth noting that this two-region behavior also appears to some extent in layers 1 and 2 before vanishing until it re-emerges at layer 17. Since a set of subsequent columns in the output projection represents the aggregated feature set from a specific attention head, the non-overlapping regions in layers 17-31 provide strong evidence for the formation of task-specialized heads alongside more general-purpose heads. This claim is further supported by the patterns in the query and key layers for this same range (17-31), where two distinct regions also exist: one with almost no overlap and another with extremely high overlap.

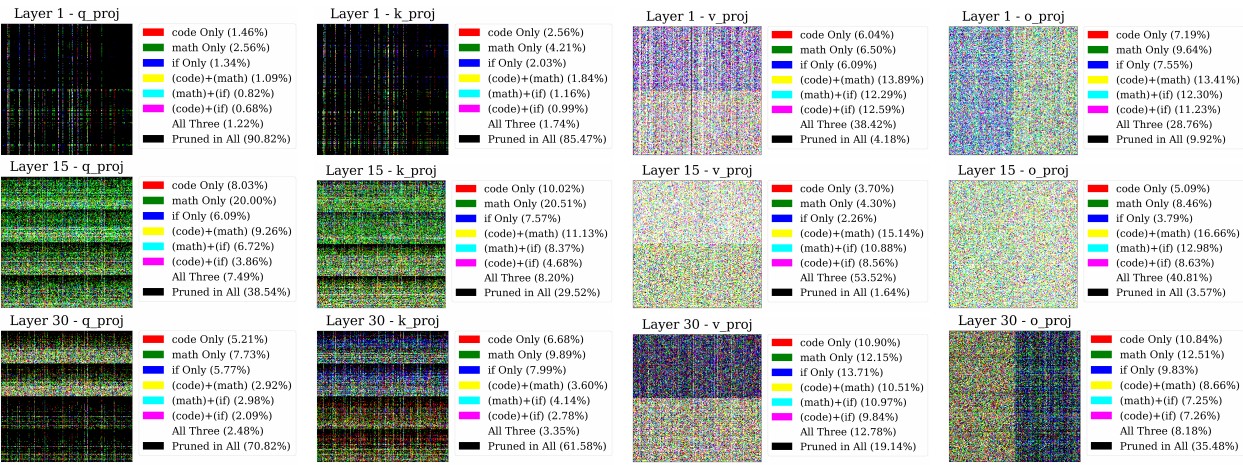

Figure 3: 3-way FFG comparison for attention components across layers 1, 15, and 30 (top to bottom). Columns show $W_q, W_k, W_v, W_o$. RGB channels represent Code (red), Math (green), and Instruction-Following (blue).

## 4.5 Analysis of Curvature and Rank Structure

**Second Moments Have Low Stable Rank, Justifying Compression.** A core assumption of our AdaFactor-style compression is that the second-moment tensors, $\mathbf{v}_\tau$, are inherently low-rank, allowing for efficient compression. We validate this by computing the stable rank of the $\mathbf{v}_\tau$ matrices for the Math and Code experts (see Figure 17 in Appendix F). Across all layers and for both SFT models, the stable rank is surprisingly low (below 1.3), confirming that the second-moment matrices are highly compressible and that a rank-1 approximation can capture a significant fraction of their energy. This provides a solid empirical justification for our memory-efficient variant of OTA, which aggressively reduces the required storage for second-moment matrices (from 29.9 GB to 12.6 MB for Llama 3.1 8b under fp32 precision) with a minor drop in model merging performance (from 0.582 to 0.571 average score), as detailed in Table 1.

**Visualizing Shared Curvature Geometry.** In this section, we leverage the connection between the second moment and diagonal curvature to study how curvature differs across SFT models. We use the same subsampling strategy for the heatmaps that was used to visualize task localization, but we apply it to the second-moment matrices instead of the grafting masks. The curvature heatmap comparisons across experts for each layer depth and weight type provide an empirical visualization suite to study our central conjecture: that SFT models fine-tuned from the same base converge to basins with highly similar curvature geometry.

In Figure 4, we visualize the log-scaled heatmaps of the square root of the `exp_avg_sq` tensor for the attention weights of two distinct SFT checkpoints, Math and Code, alongside the max-to-min ratio of their diagonal curvatures. We only report patterns that exist consistently across all layers; layer 11 is shown here as a representative example. The complete set of curvature comparison heatmaps is available in the provided anonymous GitHub repository. We first observe that the diagonal curvature has a clearly visible row-wise and column-wise structure, matching the observations in the mask visualization in Section 4.4. The column-wise

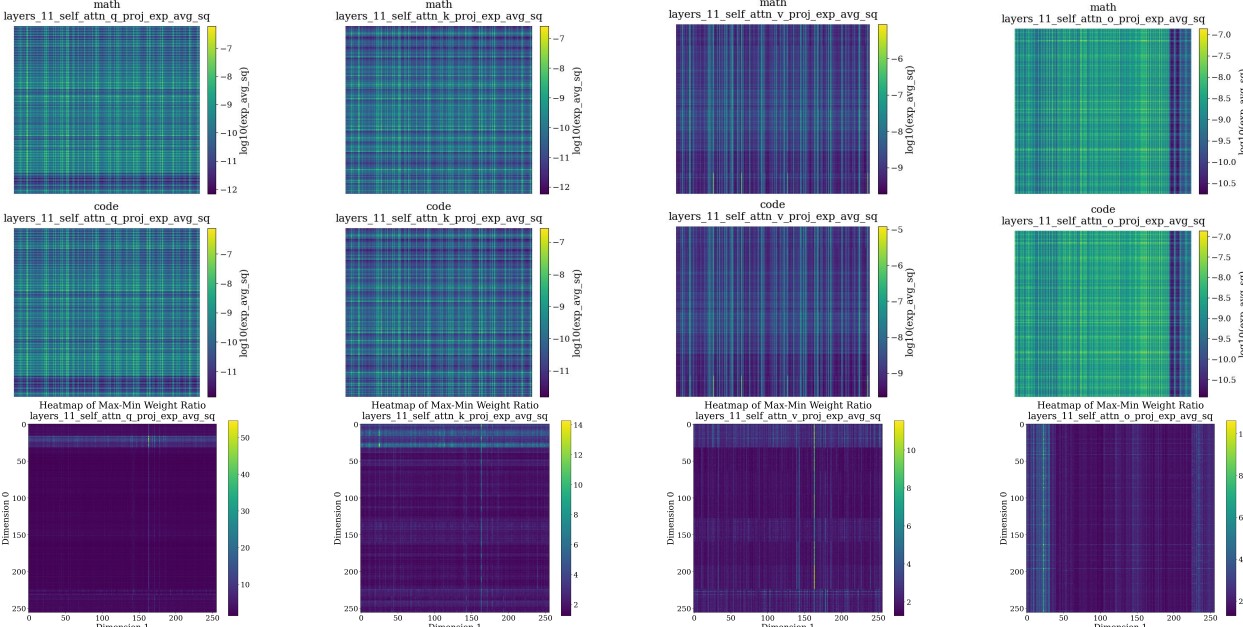

Figure 4: **Shared curvature geometry in attention layer 11.** The top two rows show the Math and Code SFT models, respectively. The striking consistency in structural patterns validates our shared curvature hypothesis. The bottom row shows the max-min ratio across all models, highlighting regions of highest variance (light color) where task specialization is most pronounced.

band can be interpreted as an **input feature curvature** for all weights connected to a given input neuron. Similarly, the row-wise band represents an **output feature curvature** for all weights connected to a given output neuron.

**How to interpret these heatmaps, and what is the takeaway?** We use our input and output curvature notions to analyze and compare curvature scales across models. We observed very high overlap in the subsets of input features with largest curvatures across all models, weight types, and layers—evident in the strong column-wise bands at identical positions for each weight type. Output features show similar high-curvature overlap. Linear max-min ratio heatmaps (Figure 4) confirm this with dark heatmaps across all layers and depths, indicating the max-min ratio remains orders of magnitude smaller than each heatmap's curvature range. For instance, layer 11's projection v-layer shows each expert with a curvature max-min ratio around $10^4$, while element-wise max-min ratios between heatmaps stay below 5 for almost all elements. Our results demonstrate significant curvature matching across models, revealing similar (though not identical) structure across training checkpoints.

This shared geometry explains linear merging's strong performance—models are already geometrically aligned, making linear merging **implicitly** curvature-aware or Fisher-optimal. While the overall structure is shared, max-min ratio plots reveal subtle but important variations in specific parameter groups where task-specific fine-tuning most significantly changed the loss landscape's curvature. This shared foundation with localized, high-variance differences motivates our two-stage approach: FFG isolates task-specific parameters before aggregating the shared structure.

To validate these differences are task-specific rather than training artifacts, we compared two Code specialists trained on identical data but with different schedulers (Cosine vs. WSD). Figure 15 in Appendix F shows virtually indistinguishable curvature patterns, quantitatively confirmed by max-min ratios an order of magnitude smaller than Code vs. Math comparisons and consistently near one. This evidence confirms our curvature-based analysis captures meaningful, task-driven geometric differences, reinforcing OTA-Merging's theoretical foundation.

## 5 Discussion

**Sparsity tuning for experts.** One of our key observations in optimizing pruning-based merging methods, such as OTA and TIES, is the necessity of tuning the sparsity ratio per expert. As shown in Figure 7, different capabilities exhibit varying sensitivity to sparsity. Some tasks, like coding, can maintain or even improve their performance under aggressive sparsity, whereas precise instruction following requires significantly lower sparsity to preserve performance. Therefore, to practically tune the sparsity per expert for each pruning-based merging method, we evaluated the corresponding capability for each SFT checkpoint at five density ratios in $\{0.05, 0.1, 0.2, 0.4, 0.6\}$ and selected the largest sparsity ratio that maintained near-full checkpoint performance. Consequently, for each capability, we performed five evaluations per merging method to conduct this tuning.

**Cost comparison: Merging vs. Data-Mixing.** One of the key goals of this paper is to benchmark capability-based model merging as an alternative to data-mixing and multi-capability training for post-training LLMs. Unlike Fisher merging, our proposed OTA method and all other baselines rely solely on lightweight element-wise matrix operations, avoiding post-hoc forward or backward passes. Consequently, the computational cost is dominated by training individual expert checkpoints and the necessary evaluations for sanity checks and sparsity tuning. For instance, training each expert requires a few days on two A100 GPUs and running the evaluation suite takes a few hours, whereas the merging process itself takes only minutes on a single A100 GPU.

From this perspective, we can meaningfully compare the compute cost of merging against the data-mixing approach used for the Tulu3 SFT checkpoint Lambert et al. (2024). A critical hyperparameter in data-mixing is the weight balance between capability datasets. Lambert et al. (2024) employed a heuristic approach, allocating a budget of five trial-and-error iterations to tune these weights. Crucially, each trial required training on the data mix from scratch followed by a full evaluation. In contrast, while we incur a similar evaluation budget for validating checkpoints and tuning sparsity, we completely eliminate the costly retraining phases. This demonstrates a significant computational advantage for model merging, particularly given the iterative nature of finding the optimal data mix.

**On the experimental scope.** Our experiments are deliberately centered on reproducing the fully open-source Tulu 3 post-training recipe Lambert et al. (2024). Unlike standard model merging studies that merge checkpoints fine-tuned on individual NLP tasks (e.g., GLUE), capability-based merging in LLM post-training demands a fully open-source workflow encompassing curated per-capability data splits, matched evaluation suites with decontamination, and validated training hyperparameters, not merely open weights. Tulu 3 is one of very few post-training recipes at the 8B scale that satisfies all of these requirements. Models such as Qwen or Mistral, while open-weight, do not publicly release the per-capability data curation, evaluation protocols, or mixing procedures needed for a controlled merging study. Our academic compute budget permitted faithful reproduction of Tulu 3 at 8B but not at its 70B or 405B variants, and designing a comparable capability-based recipe for a different model family would itself constitute a contribution on the scale of Tulu 3.

We emphasize that this controlled setup, where data, evaluations, base model, and optimizer hyperparameters are all held fixed, is a deliberate strength: it isolates merging as the sole experimental factor, which is essential for the validity of our curvature analyses, task-localization findings, and ablation studies. Moreover, we cover five of the six core capabilities defined by Tulu 3, providing near-complete coverage of the post-training skill set. Finally, the core components of OTA are architecture-agnostic, operating on generic per-parameter optimizer statistics available from any AdamW-based training run. Extending to additional architectures and scales as more fully open post-training recipes become available is a valuable direction for future work.

**Beyond diagonal approximation of the Fisher matrix.** One of the essential components of our proposed OTA framework is the direct repurposing of Adam's second moments at the end of training as an approximation of the empirical Fisher information (or curvature). However, the Adam second moment captures only the diagonal of the Fisher matrix. Consequently, the effects of off-diagonal terms on merging and pruning are not addressed in our current framework. While the full Fisher matrix is computationally intractable for LLMs, Kronecker-factored approximations have been proposed as effective non-diagonal alternatives. Optimizers such as Shampoo Gupta et al. (2018) and KFAC Martens & Grosse (2015a) leverage this concept to improve the empirical Fisher approximation and the resulting preconditioners. Inspired by

the success of these optimizers, we believe extending our OTA framework to incorporate Kronecker-based approximations is a valuable direction for future work. Moreover, even within the literature on preconditioned optimization, there is a lack of theoretical frameworks characterizing the precise benefits of non-diagonal (e.g., Kronecker-based) versus diagonal approximations. Our work currently shares this limitation. Thus, establishing the theoretical underpinnings of the trade-offs between diagonal and non-diagonal approximations remains an important objective for future research.

## 6  Conclusion

We introduced OTA Merging, a scalable and effective framework for consolidating specialized models by harnessing the rich, yet often discarded, second-moment statistics from the Adam optimizer. We demonstrated that this optimization history serves as a powerful and computationally efficient proxy for local loss landscape curvature. Our twofold approach combines FFG, which leverages curvature information as a principled denoiser to identify and revert noisy parameter updates and isolate essential knowledge from each expert, with curvature-aware aggregation that merges these denoised experts while respecting their underlying geometry. This methodology is motivated by our central discovery that independently fine-tuned models exhibit remarkable geometric consensus, shifting the primary challenge of merging from alignment to interference mitigation. Furthermore, we showed that FFG serves as a potent analytical tool, revealing structured, role-aware sparsity patterns that offer new insights into task localization. By treating the optimization trajectory as a valuable asset, OTA Merging provides a new, robust paradigm for efficient model composition, paving the way for future explorations into more complex curvature approximations and their application across different model composition techniques.

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

# A  Additional Related Works

**Pruning and Grafting with Second-Order Signals.**  Classical pruning measured parameter saliency via second-order criteria: Optimal Brain Damage (OBD) uses a diagonal Hessian approximation (LeCun et al., 1990), and Optimal Brain Surgeon (OBS) leverages full curvature (Hassibi & Stork, 1992a). Singh & Alistarh (2020) provide scalable inverse-Fisher approximations. For LLMs, one-shot/zero-shot methods such as SparseGPT (Frantar & Alistarh, 2023) and Wanda (Sun et al., 2023) enable accurate pruning without retraining; movement pruning adapts masks during fine-tuning (Sanh et al., 2020). In contrast, our objective is not generic compression: FFG computes a curvature-weighted edit saliency $s_i = H_{ii}\, \Delta w_i^2$ and grafts by resetting low-saliency coordinates to base weights. This simultaneously reduces cross-task interference and reveals interpretable task localization.

**Mode Connectivity and Alignment.**  Mode-connectivity work shows that the independently trained checkpoints on same data are often connected by low-loss paths (Garipov et al., 2018). After permutation alignment, independently trained networks lie in an approximately convex basin, which explains why linear interpolation/merging can work when models are geometrically aligned (Ainsworth et al., 2023). Our curvature-aware view complements these results: if diagonal curvature morphology is shared across specialists, then linear aggregation with curvature reweighting is particularly effective.

**Comparison with Li et al. (2025)**  While Li et al. (2025) study Fisher pruning applied to the final model weights ($\mathbf{w}^*$) by setting parameters to zero, FFG instead operates on the task vector ($\Delta \mathbf{w} = \mathbf{w}^* - \mathbf{w}_0$) to revert low-saliency updates. This mitigates interference between non-IID experts—a critical step for our SFT merging setting. This denoising role is the cornerstone of our OTA-Merging framework and is a key differentiator from other merging methods. Second, we uniquely employ the second moment proxy as an analytical and interpretability lens. We use it to propose and provide strong empirical evidence for a shared curvature hypothesis, offering a new explanation for the effectiveness of model merging. Furthermore, we leverage FFG as a tool for task localization to understand SFT training regimes, revealing how skills are encoded via structured, role-aware sparsity patterns in the network, a line of analysis not pursued in Li et al. (2025).

Finally, to address the significant practical issue of storage, we propose and validate an AdaFactor-style rank-1 compression of the second-moment tensor. This reduces the storage overhead significantly, making our approach highly scalable for large models. In summary, while Li et al. (2025) establish the broad utility of the optimizer-as-Fisher proxy, our work presents a specialized, end-to-end framework, and benchmarks for the challenging SFT merging problem at a **non-trivial scale**, completed with a novel denoising mechanism, new interpretability insights, and a practical, scalable implementation.

# B  Preliminaries

This section establishes the notation and foundational concepts that underpin our work. We begin by formalizing the SFT setup and the associated notation. We then introduce the Fisher Information Matrix (FIM) as a key tool for understanding the curvature of the loss landscape. Finally, we review how second-order information, approximated by the FIM, is leveraged in established methods for model merging and parameter grafting, setting the stage for our proposed contributions.

## B.1  SFT Setup

**Notation.** We denote matrices with bold capital letters ($\mathbf{A}$), vectors with bold lowercase letters ($\mathbf{v}$), and scalars with regular lowercase letters ($s$). A vector-valued function's $j^{\text{th}}$ output is denoted as $f^j$. The $i$-th standard basis vector is $\mathbf{e}_i$, and an $n$-dimensional vector of ones is $\mathbf{1}_n$. For any Positive Semi-Definite (PSD) matrix $\mathbf{P} \in \mathbb{R}^{d \times d}$, we define its induced norm on a vector $\mathbf{x} \in \mathbb{R}^d$ as $\|\mathbf{x}\|_{\mathbf{P}} = \sqrt{\mathbf{x}^\top \mathbf{P} \mathbf{x}}$.

**Learning Setup.** We consider a supervised fine-tuning (SFT) scenario with $T$ distinct tasks from the same base model. For each task $\tau \in \{1, \ldots, T\}$, we have a dataset $\mathcal{S}_\tau = \{(\mathbf{x}_i^\tau, y_i^\tau)\}_{i=1}^{|\mathcal{S}_\tau|}$, with samples drawn from

a true data distribution $D_\tau$. We begin with a common pre-trained model architecture, parameterized by $\boldsymbol{w}_0 \in \mathbb{R}^d$, which is then fine-tuned for each specific task. The model, $f(\cdot; \boldsymbol{w})$, maps an input $\mathbf{x} \in \mathcal{X}$ to a logit vector $\mathbf{z} \in \mathbb{R}^{|V|}$, where $|V|$ is the vocabulary size. These logits parameterize a conditional probability distribution $P(y|\mathbf{x}; \boldsymbol{w})$ via the softmax function: $\log p(y|\mathbf{x}; \boldsymbol{w}) = \mathbf{e}_y^\top \mathbf{z} - \log\left(\sum_{j=1}^{|V|} \exp(z_j)\right)$.

The objective for each task $\tau$ is to minimize the empirical cross-entropy loss, which approximates the true expected risk over the data distribution $D_\tau$:

$$\mathcal{L}_{\mathcal{S}_\tau}(\boldsymbol{w}) = -\mathbb{E}_{(\mathbf{x},y)\sim\mathcal{S}_\tau}\left[\log p(y|\mathbf{x}; \boldsymbol{w})\right].$$

For notational simplicity, we will drop the task subscript $\tau$ when the context is clear.

## B.2 The Fisher Information Matrix

A central concept for analyzing the loss landscape is the Fisher Information Matrix (FIM), which measures the sensitivity of the model's output distribution to changes in its parameters $\boldsymbol{w}$. It provides a powerful approximation of the loss curvature Amari (1998); Martens (2020) and is equivalent to the negative Hessian of the log-likelihood, under expectation over model's predictive distributions, $\mathbf{F}(\boldsymbol{w}) = -\mathbb{E}_{y\sim P(y|\boldsymbol{x};\boldsymbol{w})}[\nabla_{\boldsymbol{w}}^2 \log p(y \mid \mathbf{x}; \boldsymbol{w})]$. In practice, several variants of the FIM are used:

**True FIM**, $\mathbf{F}(\boldsymbol{w})$, is defined over the true data distribution and the model's predictive distribution, making it intractable for deep neural networks:

$$\mathbf{F}(\boldsymbol{w}) = \mathbb{E}_{\mathbf{x}\sim D(\mathbf{x}),y\sim P(y|\mathbf{x};\boldsymbol{w})}\left[\nabla_{\boldsymbol{w}} \log p(y|\mathbf{x}; \boldsymbol{w})\nabla_{\boldsymbol{w}} \log p(y|\mathbf{x}; \boldsymbol{w})^\top\right]. \tag{2}$$

**Expected Empirical FIM**, $\hat{\mathbf{F}}(\boldsymbol{w})$, approximates the true FIM by using a finite dataset $\mathcal{S}$ but still requires an expectation over the model's predictions:

$$\hat{\mathbf{F}}(\boldsymbol{w}) = \frac{1}{|\mathcal{S}|} \sum_{(\mathbf{x}_i,y_i)\in\mathcal{S}} \mathbb{E}_{y\sim P(y|\mathbf{x}_i;\boldsymbol{w})}\left[\nabla_{\boldsymbol{w}} \log p(y|\mathbf{x}_i; \boldsymbol{w})\nabla_{\boldsymbol{w}} \log p(y|\mathbf{x}_i; \boldsymbol{w})^\top\right]. \tag{3}$$

**Observed Empirical FIM**, $\bar{\mathbf{F}}(\boldsymbol{w})$, simplifies this further by replacing the expectation with the observed ground-truth labels from the dataset. This variant, often called the "empirical Fisher," is the most commonly used in practice Martens & Grosse (2015b); Matena & Raffel (2022b) due to its computational advantage by avoiding the need for costly sampling from model's distribution:

$$\bar{\mathbf{F}}(\boldsymbol{w}) = \frac{1}{|\mathcal{S}|} \sum_{(\mathbf{x}_i,y_i)\in\mathcal{S}} \left[\nabla_{\boldsymbol{w}} \log p(y_i|\mathbf{x}_i; \boldsymbol{w})\nabla_{\boldsymbol{w}} \log p(y_i|\mathbf{x}_i; \boldsymbol{w})^\top\right]. \tag{4}$$

## B.3 Applications of Second-Order Information

The FIM's ability to capture loss curvature makes it invaluable for a range of model manipulation techniques, from merging diverse experts to compressing a single model.

**Preconditioned Model Merging.** The goal of model merging is to combine a set of fine-tuned expert models, $\{\boldsymbol{w}_\tau^*\}_{\tau=1}^T$, into a single, multi-tasked model $\boldsymbol{w}_{\text{merged}}$. A generalized approach involves a weighted average in parameter space Tam et al. (2023):

$$\boldsymbol{w}_{\text{merged}} = \left(\sum_{\tau=1}^T \mathbf{C}_\tau\right)^{-1} \left(\sum_{\tau=1}^T \mathbf{C}_\tau \boldsymbol{w}_\tau^*\right), \tag{5}$$

where $\mathbf{C}_\tau$ are PSD weighting matrices. Different choices for $\mathbf{C}_\tau$ yield different merging strategies. For instance, Fisher-weighted averaging Matena & Raffel (2022b) uses the empirical FIM of each task as $\mathbf{C}_\tau$, leveraging the loss landscape geometry to guide the combination process, and Tam et al. (2023) leveraged Kronecker factored approximation of empirical fisher (Martens & Grosse, 2015a) for the choice of $\mathbf{C}_\tau$.

**Optimal Brain Damage for Pruning and Grafting.** Second-order information is also fundamental to classic model compression techniques like Optimal Brain Damage (OBD) LeCun et al. (1989); Hassibi & Stork (1992b). OBD identifies and removes parameters with the smallest impact on the loss function. This impact, or "saliency," is estimated via a second-order Taylor expansion. For a model at a local minimum $\boldsymbol{w}^*$, a small parameter perturbation $\delta\boldsymbol{w}$ changes the loss by $\Delta\mathcal{L}_\mathcal{S} \approx \frac{1}{2}(\delta\boldsymbol{w})^\top \mathbf{H}(\delta\boldsymbol{w})$, where $\mathbf{H}$ is the Hessian, approximated by the FIM. To make this tractable, OBD typically uses only the diagonal of the Hessian. Pruning a parameter $w_i^*$ to zero corresponds to a saliency score of $s_i = \frac{1}{2}\mathbf{H}_{ii}(w_i^*)^2$.

This framework can be repurposed from pruning to **grafting**. Instead of nullifying parameters, we can selectively revert fine-tuned parameters $\boldsymbol{w}^*$ back to their pre-trained state $\boldsymbol{w}^0$. The perturbation becomes $\delta w_i = w_i^* - w_i^0$, and the saliency of keeping the fine-tuned update is calculated as $s_i = \frac{1}{2}\mathbf{H}_{ii}(w_i^* - w_i^0)^2$. This score quantifies the importance of the change acquired during fine-tuning, providing a direct link to merging by deciding on a parameter-wise basis whether to retain a specialized update or revert to the base model.

## B.4 Efficiently Estimating Second-Order Information

A major challenge in using second-order methods is the formidable memory cost of storing the full FIM or Hessian. While generic low-rank approximations like SVD exist, they do not guarantee the preservation of non-negativity, a defining property of these matrices.

**Factored Estimators (AdaFactor).** The AdaFactor optimizer Shazeer & Stern (2018) introduces a memory-efficient factorization that guarantees non-negativity. For a matrix of squared-gradient Exponential Moving Averages (EMAs) $\mathbf{V} \in \mathbb{R}^{m \times n}$, AdaFactor avoids storing the full $mn$ elements. Instead, it maintains only the moving averages of its row and column sums: $\mathbf{r} = \mathbf{V}\mathbf{1}_n \in \mathbb{R}^m$ and $\mathbf{c}^\top = \mathbf{1}_m^\top \mathbf{V} \in \mathbb{R}^{1 \times n}$. A rank-1, non-negative approximation of the full matrix is then reconstructed as $\hat{\mathbf{V}} = \mathbf{r}\mathbf{c}^\top/(\mathbf{1}_m^\top \mathbf{r})$. This reduces storage from $O(mn)$ to $O(m+n)$ per parameter matrix. The effectiveness of such low-rank approximations is often justified by the concept of stable rank, $r_s(\mathbf{V}) = \|\mathbf{V}\|_F^2/\|\mathbf{V}\|_2^2$, which measures how well a matrix can be approximated by a low-rank counterpart. While AdaFactor was designed to save memory during training, we propose leveraging its factorization after training to create a highly compressed snapshot of the second-moment matrix, providing nearly storage-free access to valuable curvature information.

# C  Theoretical Justifications and Proofs

This section provides a detailed derivation of the theoretical insights on Adam's second-moment accumulator, $\mathbf{v}$, as a principled proxy for the diagonal of the empirical Fisher Information Matrix (FIM).

**Theoretical Justification.** The core argument rests on the equivalence between the Hessian of the loss function ($\nabla^2\mathcal{L}_D$) and the Observed Empirical FIM ($\bar{\mathbf{F}}$) near a local minimum $\mathbf{w}_*$.

1. We first assume the network's output is locally linear with respect to its parameters near the end of training (the **Late NTK Regime**). This allows us to approximate the Hessian with the Generalized Gauss-Newton (GGN) matrix. While neural network training has been shown not to be well-approximated by the NTK at initialization, meaning an aggressive kernel change is necessary for feature learning (Vyas et al., 2022), the kernel has been shown to stabilize near the end of training (Fort et al., 2020).

2. We then assume the model is well-calibrated at convergence (**Perfect Calibration**), meaning its predictive distribution matches the true data distribution.

Under these assumptions, one can argue that the Hessian is approximately equal to the Observed Empirical FIM: $\nabla^2\mathcal{L}_D(\mathbf{w}_*) \approx \bar{\mathbf{F}}(\mathbf{w}_*)$. Furthermore, the expectation of the outer product of mini-batch gradients is a scaled version of the FIM: $\mathbb{E}_{B_k \sim D^{|B|}}[\nabla\mathcal{L}_{B_k}\nabla\mathcal{L}_{B_k}^\top] = \frac{1}{|B|}\bar{\mathbf{F}}(\mathbf{w}_*)$. We would like to highlight that the equivalence of the Empirical Fisher information with the Hessian under perfect calibration is a well-known property Amari (1998), and the expectation of the mini-batch gradient outer product is a corollary of Lemma 8 in Morwani et al. (2024).

Thus here, we provide a strong theoretical argument for our method: the Adam second-moment accumulator, $\mathbf{v}$, on expectation, is a scaled EMA of the diagonal of the FIM. It is therefore a valid and computationally free proxy for the diagonal curvature of the loss landscape, which we can harness for both parameter selection and model merging.

### C.1 Proof of Equivalence between Hessian and Empirical FIM

Our first result connects the Hessian of the loss function to the Observed Empirical FIM at a fine-tuned model's optimal parameters, $\mathbf{w}_*$. The Hessian for a loss $\mathcal{L}_D(\mathbf{w})$ over a dataset $D$ is given by:

$$\nabla^2 \mathcal{L}_D(\mathbf{w}) = \mathbb{E}_{(\mathbf{x},y)\sim D}\left[\underbrace{\mathbf{J}_f(\mathbf{w},\mathbf{x})^\top \nabla_\mathbf{z}^2 \ell(y,\mathbf{z})\mathbf{J}_f(\mathbf{w},\mathbf{x})}_{\text{Generalized Gauss-Newton (GGN) term}}\right] + \mathbb{E}_{(\mathbf{x},y)\sim D}\left[\underbrace{\sum_{j=1}^{C}\left(\nabla_{z^j}\ell(y,\mathbf{z})\right)\nabla_\mathbf{w}^2 f^j(\mathbf{w},\mathbf{x})}_{\text{Second-order term}}\right], \quad (6)$$

where $\mathbf{z} = f(\mathbf{w},\mathbf{x})$ are the model's logits, $\ell$ is the per-sample loss (e.g., negative log-likelihood), and $\mathbf{J}_f$ is the Jacobian of the network function $f$ with respect to the parameters $\mathbf{w}$. To simplify this expression, we rely on two standard assumptions.

**Assumption 1** (Late NTK Locality). *Near an optimal set of parameters $\mathbf{w}_*$, the second-order term in Equation (6), which depends on the curvature of the network function $f$ itself, is negligible. This implies local linearity: $f(\mathbf{x};\mathbf{w}_* + \boldsymbol{\delta}) \approx f(\mathbf{x};\mathbf{w}_*) + \mathbf{J}_f\boldsymbol{\delta}$ for small perturbations $\boldsymbol{\delta}$.*

This assumption allows us to approximate the Hessian using only the GGN term:

$$\nabla^2 \mathcal{L}_D(\mathbf{w}_*) \approx \mathbb{E}_{(\mathbf{x},y)\sim D}\left[\mathbf{J}_f(\mathbf{w}_*,\mathbf{x})^\top \nabla_\mathbf{z}^2 \ell(y,f(\mathbf{x};\mathbf{w}_*))\mathbf{J}_f(\mathbf{w}_*,\mathbf{x})\right]. \quad (7)$$

**Assumption 2** (Perfect Calibration at Fine-Tuned Checkpoints). *At the optimal parameters $\mathbf{w}_*$, the model is perfectly calibrated, meaning its predictive distribution matches the true conditional data distribution for any given input $\mathbf{x}$: $p(y|\mathbf{x};\mathbf{w}_*) = d(y|\mathbf{x})$.*

With these assumptions, we can now state and prove the main lemma.

**Lemma 1.** *Under late NTK locality (Assumption 1), the Hessian of the loss $\nabla^2 \mathcal{L}_D(\mathbf{w}_*)$ can be decomposed as follows:*

$$\begin{aligned}
\nabla^2 \mathcal{L}_D(\mathbf{w}_*) = &\underbrace{\mathbb{E}_{(\mathbf{x},y)\sim D}\left[\nabla\mathcal{L}_D(\mathbf{w}_*,\mathbf{x},y)\nabla\mathcal{L}_D(\mathbf{w}_*,\mathbf{x},y)^\top\right]}_{\text{Empirical FIM}} \\
&- \underbrace{\mathbb{E}_{\mathbf{x}\sim D_\mathbf{x}}\left[\mathbb{E}_{y\sim D_{y|\mathbf{x}}}\left[\nabla\mathcal{L}_D(\mathbf{w}_*,\mathbf{x},y)\right]\mathbb{E}_{y\sim D_{y|\mathbf{x}}}\left[\nabla\mathcal{L}_D(\mathbf{w}_*,\mathbf{x},y)\right]^\top\right]}_{\text{Expected Gradient Covariance}} \\
&+ \underbrace{\mathbb{E}_{\mathbf{x}\sim D_\mathbf{x}}\left[\mathbf{J}_f(\mathbf{w}_*,\mathbf{x})^\top\left(\boldsymbol{\Sigma}_p - \boldsymbol{\Sigma}_d\right)\mathbf{J}_f(\mathbf{w}_*,\mathbf{x})\right]}_{\text{Covariance Mismatch}}
\end{aligned} \quad (8)$$

*where $\boldsymbol{\Sigma}_p = Cov_{y\sim p(\cdot|f(\mathbf{x};\mathbf{w}_*))}[\mathbf{e}_y]$, $\boldsymbol{\Sigma}_d = Cov_{y\sim D_{y|\mathbf{x}}}[\mathbf{e}_y]$, and $\mathbf{e}_y$ is a one-hot vector representing label $y$. Moreover, $\nabla\mathcal{L}_D(\mathbf{w}_*,\mathbf{x},y) = \nabla_\mathbf{w} \log p(y|\mathbf{x};\mathbf{w}_*)$ is the log-probability gradient at a given sample $(\mathbf{x},y)$. $D_{y|\mathbf{x}}$ and $p(\cdot|f(\mathbf{x};\mathbf{w}_*))$ are the true conditional distribution and the model distribution given input $\mathbf{x}$, respectively. $D_\mathbf{x}$ denotes the true marginal distribution.*

**Remark.** The above lemma characterizes the relation between the empirical Fisher and the Hessian under the late NTK locality assumption. Interestingly, we observe that the approximation errors consist of the covariance mismatch between the model's prediction and the true distribution, as well as the expected gradient covariance term. Under perfect calibration (Assumption 2), it is well-known that the Hessian would be equivalent to the empirical FIM Amari (1998). While the perfect calibration assumption is reasonable at the convergence point $\mathbf{w}_*$, our proposed Lemma 1 does not depend on perfect calibration and establishes the equivalence relation only under late NTK locality. Moreover, note that both the expected gradient over labels and the covariance mismatch terms in Equation 8 would be equal to zero under perfect calibration.

*Proof.* Given input $\mathbf{x}$, let $\mathbf{p}(\cdot|\mathbf{x})$ and $\mathbf{d}(\cdot|\mathbf{x})$ denote the probability vectors for the model and the true distribution. For ease of notation, we refer to these vectors as $\mathbf{p}$ and $\mathbf{d}$. We begin by analyzing the inner term of the GGN in Equation (7), which is the expected Hessian of the negative log-likelihood $\nabla_{\mathbf{z}}^2 \ell(y, \mathbf{z}) = \nabla^2 \log p(y|\mathbf{z})$, where $\mathbf{z} = f(\mathbf{x}; \mathbf{w}_*)$. For the softmax cross-entropy loss on logits, $\log p(y|\mathbf{z}) = z_y - \log(\sum_{i=1}^{|V|} e^{z_i})$, we can write:

$$\nabla_{\mathbf{z}}^2 \ell(y, \mathbf{z}) = \text{diag}(\mathbf{p}) - \mathbf{p}\mathbf{p}^\top = \mathbf{\Sigma}_p$$
$$\nabla_{\mathbf{z}} \ell(y, \mathbf{z}) = \mathbf{e}_y - \mathbf{p} \tag{9}$$

Now let $\bar{F}_{\mathbf{z}} = \mathbb{E}_{y \sim D_{y|\mathbf{x}}}[\nabla_{\mathbf{z}}\ell(y, \mathbf{z})\nabla_{\mathbf{z}}\ell(y, \mathbf{z})^\top] = \mathbb{E}_{y \sim \mathbf{d}(\cdot|\mathbf{x})}[(\mathbf{e}_y - \mathbf{p})(\mathbf{e}_y - \mathbf{p})^\top]$. Expanding the expectation, we have:

$$\bar{F}_{\mathbf{z}} = \text{diag}(\mathbf{d}) - \mathbf{d}\mathbf{p}^\top - \mathbf{p}\mathbf{d}^\top + \mathbf{p}\mathbf{p}^\top = \text{diag}(\mathbf{d}) - \mathbf{d}\mathbf{d}^\top + (\mathbf{d} - \mathbf{p})(\mathbf{d} - \mathbf{p})^\top \tag{10}$$

Hence, we can write: $\nabla_{\mathbf{z}}^2 \ell(y, \mathbf{z}) - \bar{F}_{\mathbf{z}} = (\mathbf{\Sigma}_p - \mathbf{\Sigma}_d) - (\mathbf{d} - \mathbf{p})(\mathbf{d} - \mathbf{p})^\top$. Combining this equation with the fact that $\mathbb{E}_{y \sim D_{y|\mathbf{x}}}[\nabla_{\mathbf{z}}\ell(y, \mathbf{z})] = \mathbf{d} - \mathbf{p}$, and $\nabla_{\mathbf{w}} \log p(y|\mathbf{x}; \mathbf{w}) = \mathbf{J}_f(\mathbf{w}, \mathbf{x})^\top \nabla_{\mathbf{z}}\ell(y, \mathbf{z})$, we can rewrite the GGN in Equation (7) to arrive at Equation (8). □

## C.2 Proof of Relation between Mini-Batch Second Moment and FIM

Next, we show how the second moment of mini-batch gradients relates to the Empirical FIM defined above.

**Lemma 2.** *Let* $\nabla\mathcal{L}_{B_k}(\mathbf{w}) = \frac{1}{|B|}\sum_{(\mathbf{x},y)\in B_k} \nabla\mathcal{L}_{\mathbf{x},y}(\mathbf{w})$ *be the gradient for a mini-batch* $B_k$ *of size* $|B|$ *sampled from the data distribution $D$. Under Assumption 2, the expectation of the outer product of this mini-batch gradient is a scaled version of the FIM:*

$$\mathbb{E}_{B_k \sim D^{|B|}} \left[ \nabla\mathcal{L}_{B_k}(\mathbf{w}_*)\nabla\mathcal{L}_{B_k}(\mathbf{w}_*)^\top \right] = \frac{1}{|B|}\bar{\mathbf{F}}(\mathbf{w}_*). \tag{11}$$

*Proof.* We decompose the expectation of the outer product:

$$\mathbb{E}_{B_k \sim D^{|B|}} \left[ \nabla\mathcal{L}_{B_k}\nabla\mathcal{L}_{B_k}^\top \right] = \mathbb{E}\left[ \left( \frac{1}{|B|}\sum_{i=1}^{|B|}\nabla\mathcal{L}_{\mathbf{x}_i,y_i} \right)\left( \frac{1}{|B|}\sum_{j=1}^{|B|}\nabla\mathcal{L}_{\mathbf{x}_j,y_j} \right)^\top \right]$$

$$= \frac{1}{|B|^2}\sum_{i,j}\mathbb{E}\left[ \nabla\mathcal{L}_{\mathbf{x}_i,y_i}\nabla\mathcal{L}_{\mathbf{x}_j,y_j}^\top \right]$$

$$= \frac{1}{|B|^2}\sum_{i=1}^{|B|}\mathbb{E}_{\mathbf{x}_i,y_i \sim D}[\nabla\mathcal{L}_{\mathbf{x}_i,y_i}\nabla\mathcal{L}_{\mathbf{x}_i,y_i}^\top] + \frac{1}{|B|^2}\sum_{i \neq j}\mathbb{E}\left[ \nabla\mathcal{L}_{\mathbf{x}_i,y_i}\nabla\mathcal{L}_{\mathbf{x}_j,y_j}^\top \right].$$

Since the samples in the mini-batch are i.i.d., the expectation of the cross-terms $(i \neq j)$ decouples:

$$\mathbb{E}\left[ \nabla\mathcal{L}_{\mathbf{x}_i,y_i}\nabla\mathcal{L}_{\mathbf{x}_j,y_j}^\top \right] = \mathbb{E}_{\mathbf{x}_i,y_i \sim D}[\nabla\mathcal{L}_{\mathbf{x}_i,y_i}]\mathbb{E}_{\mathbf{x}_j,y_j \sim D}[\nabla\mathcal{L}_{\mathbf{x}_j,y_j}]^\top.$$

At the optimum $\mathbf{w}_*$, the expected gradient over the true data distribution is zero. This follows from Assumption 2 and the fact that the expectation of the score function is zero:

$$\mathbb{E}_{(\mathbf{x},y) \sim D}\left[ \nabla_{\mathbf{w}}\mathcal{L}_{\mathbf{x},y}(\mathbf{w}_*) \right] = \mathbb{E}_{\mathbf{x}}\left[ \mathbf{J}_f^\top \mathbb{E}_{y \sim p(y|\mathbf{x})}[\nabla_{\mathbf{z}} \log p(y|\mathbf{z})] \right] = \mathbf{0}.$$

Therefore, all the cross-terms $(i \neq j)$ in the decomposition vanish. We are left with only the diagonal terms of the sum:

$$\mathbb{E}_{B_k \sim D^{|B|}} \left[ \nabla\mathcal{L}_{B_k}\nabla\mathcal{L}_{B_k}^\top \right] = \frac{1}{|B|^2}\sum_{i=1}^{|B|}\mathbb{E}\left[ \nabla\mathcal{L}_{\mathbf{x}_i,y_i}\nabla\mathcal{L}_{\mathbf{x}_i,y_i}^\top \right]$$

$$= \frac{|B|}{|B|^2}\mathbb{E}_{(\mathbf{x},y) \sim D}\left[ \nabla\mathcal{L}_{\mathbf{x},y}(\mathbf{w}_*)\nabla\mathcal{L}_{\mathbf{x},y}(\mathbf{w}_*)^\top \right]$$

$$= \frac{1}{|B|}\bar{\mathbf{F}}(\mathbf{w}_*).$$

This completes the proof, showing that the second moment of the mini-batch gradient is, on expectation, a scaled version of the full Empirical FIM. □

# D   Additional Information on Training SFT Models

## D.1   Gradient, and Loss Visualizations

Figure 5 shows the training gradient norm trajectories $\|\nabla\mathcal{L}\|$ for each of our five specialist LLMs. Tasks like `knowledge_recall` and `coding` exhibit high gradient norms, whereas `math_reasoning` and `precise_if` remain much lower.

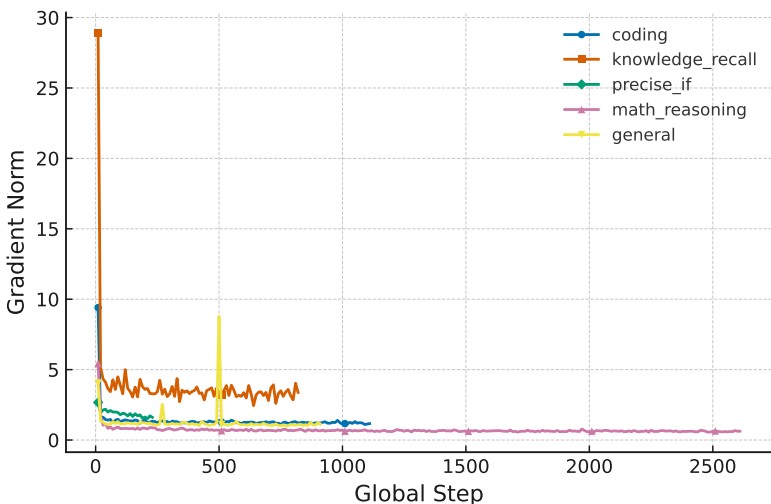

Figure 5: Gradient norm trajectories over global training steps for each fine-tuned SFT models

Moreover, we visualize the optimization trajectories of each SFT model on distinct tasks as indicated in Figure 6.

## D.2   Experiments Details for Training SFT Checkpoints

The models were trained from the `Meta-Llama-3.1-8B` base model using full-parameter Supervised Fine-Tuning (SFT) and AdamW optimizer. The second-moment statistics (`exp_avg_sq`) of the optimizer were checkpointed and later used as curvature proxies in OTA-Merging.

We leveraged full post-training stack provided by **LLaMA-Factory** Zheng et al. (2024). This unified infrastructure handled supervised fine-tuning (SFT), tokenizer alignment, and checkpoint conversion.

Our configuration closely followed the hyperparameter recipe from the Tülu-3 Lambert et al. (2024), with slight task-specific adjustments. Fine-tuning was performed on two NVIDIA A100 GPUs (80GB) per task. We used gradient accumulation of 32 and a micro-batch size of 2 per device, yielding an effective global batch size of 128. All models were trained with a learning rate of $5 \times 10^{-6}$, except for Precise Instruction Following (IF), which used $1 \times 10^{-5}$ to encourage faster convergence.

## D.3   Training Dataset Curation

Our SFT models were fine-tuned on curated subsets of the `allenai/tulu-3-sft-mixture` dataset, retrieved from the Hugging Face `datasets` library. This dataset aggregates instruction-following conversations from a wide range of sources, and we mapped these sources to specific capability categories for training the OTA-Merging specialists.

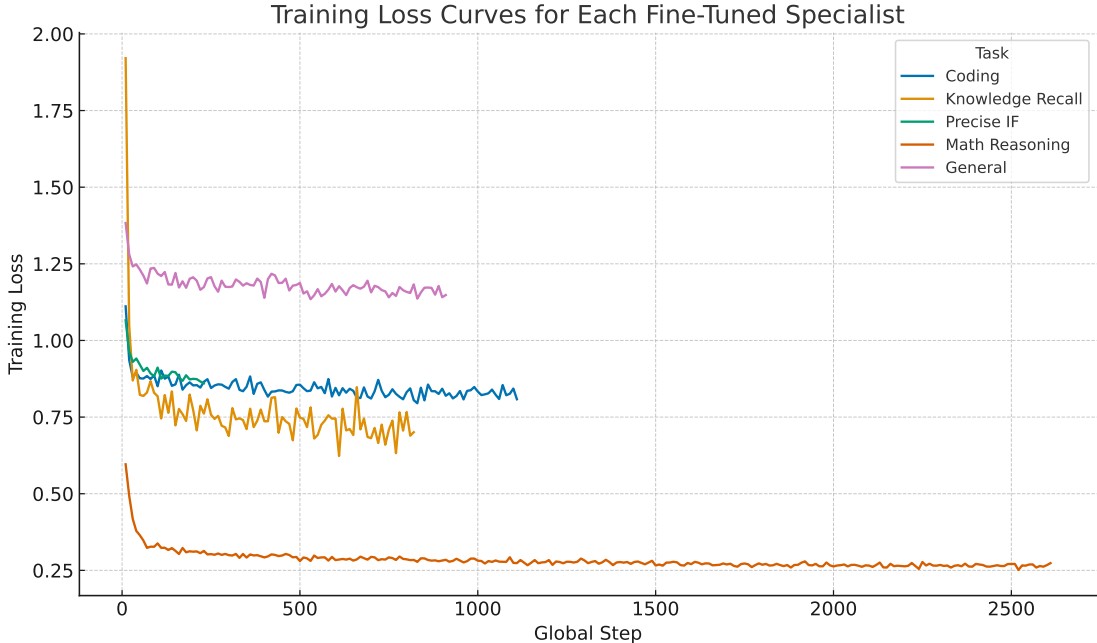

Figure 6: Training loss trajectories across global steps for each SFT model

Table 2: Fine-tuning hyperparameters used for training specialist models.

| Hyperparameter | Value | Notes |
|---|---|---|
| Base Model | LLaMA 3.1–8B | |
| Micro Batch Size | 2 | Per GPU |
| Gradient Accumulation | 32 | |
| Effective Batch Size | 128 | Across GPUs |
| Max Token Length | 4096 | |
| Learning Rate | $5 \times 10^{-6}$ | $1 \times 10^{-5}$ for Precise IF |
| Learning Rate Schedule | Linear | |
| Warmup Ratio | 0.03 | |
| Epochs | 1 | |
| Post-training Stack | LLaMA-Factory | Full stack used Zheng et al. (2024) |

Each training example consists of a structured conversation in the `messages` field—a list of dictionaries that capture the dialogue turns between roles: `system`, `user`, and `assistant`. This structure enables role-specific formatting for instruction tuning.

**Task Categories and Source Mapping.** The data sources used for fine-tuning were grouped into the following categories:

- **General Instruction**: `wildchat`, `oasst1_converted`, `no_robots`, etc.

- **Knowledge Recall**: `flan_v2`, `sciriff`, `table_gpt`

- **Mathematical Reasoning**: `persona_math`, `numinamath`, `open_math`

- **Coding**: `codealpaca`, `persona_code`

- **Precise Instruction Following (Precise IF)**: `persona_ifdata`

**Dataset Statistics.** The number of examples in each category used for fine-tuning is summarized below:

Table 3: Number of examples per task category in `tulu-3-sft-mixture`.

| Category | Examples | Percentage |
|---|---|---|
| Mathematical Reasoning | 334,252 | 39.70% |
| Coding | 142,275 | 16.89% |
| General Instruction | 116,871 | 13.89% |
| Knowledge Recall | 104,982 | 12.46% |
| Precise Instruction Following | 29,980 | 3.56% |
| **Total** | 728,360 | 100.00% |

**Message Formatting.** Each `messages` list was rendered into a flattened training string using a standardized Jinja2 template consistent with LLaMA-Factory's post-training stack. This template inserts role-specific delimiters and appends `<eos_token>` after assistant responses.

For example, the following JSON input:

```
[
  {"role": "system", "content": "System prompt."},
  {"role": "user", "content": "User question."},
  {"role": "assistant", "content": "Assistant answer."}
]
```

is rendered as:

```
<|system|>
System prompt.
<|user|>
User question.
<|assistant|>
Assistant answer.<eos_token>
```

This formatting ensures consistency across training samples and compatibility with instruction-tuned decoding patterns adopted from Tulu Lambert et al. (2024).

### D.4  SFT Checkpoints Evaluations

**SFT Models Localize Distinct Capabilities.** First, we establish a baseline by analyzing the performance of individual SFT models (Table 4). As expected, each expert excels on benchmarks aligned with its training data: the Math specialist outperforms all others on `MATH` (0.316), and the Coding specialist dominates `HumanEval` (0.788). Conversely, the near-zero scores of non-coding specialists on `HumanEval` underscore the non-IID nature of the training data and highlight the core challenge of merging: combining these complementary but isolated skills without destructive interference. Moreover, the SFT experts appear to either outperform or closely match the performance of the Tulu-3 SFT checkpoint (the multi-task SFT tuned model), with the exception of coding capability. For coding, the Tulu models have considerably higher performance on HumanEval and HumanEval+ compared to the SFT coding model. This suggests that this task benefits the most from multi-task learning, as code-only filtered subset of the Tulu SFT mixture was unable to retain the performance of the multi-instructed models.

Table 4: Performance of individual SFT experts and the multi-task Tulu-3-8B SFT model on their respective benchmarks. Best performance per row is shown in bold. Scores are reported in $[0, 1]$.

| Benchmark | Specialists | | | | | Tulu-3 SFT |
|---|---|---|---|---|---|---|
| | General | Knowledge | Math | Precise IF | Coding | |
| BBH-CoT | 0.671 | 0.634 | 0.650 | 0.628 | 0.635 | **0.688** |
| HumanEval | 0.000 | 0.000 | 0.700 | 0.688 | 0.788 | **0.866** |
| HumanEval+ | 0.000 | 0.000 | 0.659 | 0.632 | 0.744 | **0.805** |
| DROP | 0.571 | **0.629** | 0.583 | 0.586 | 0.552 | 0.616 |
| GSM8K | 0.575 | 0.589 | 0.757 | 0.594 | 0.561 | **0.767** |
| IFEval | 0.516 | 0.538 | 0.257 | **0.717** | 0.425 | 0.715 |
| MATH | 0.170 | 0.171 | **0.316** | 0.199 | 0.181 | 0.290 |
| POPQA | **0.317** | 0.301 | 0.296 | 0.307 | 0.301 | 0.295 |

# E   Complementary FFG Analysis

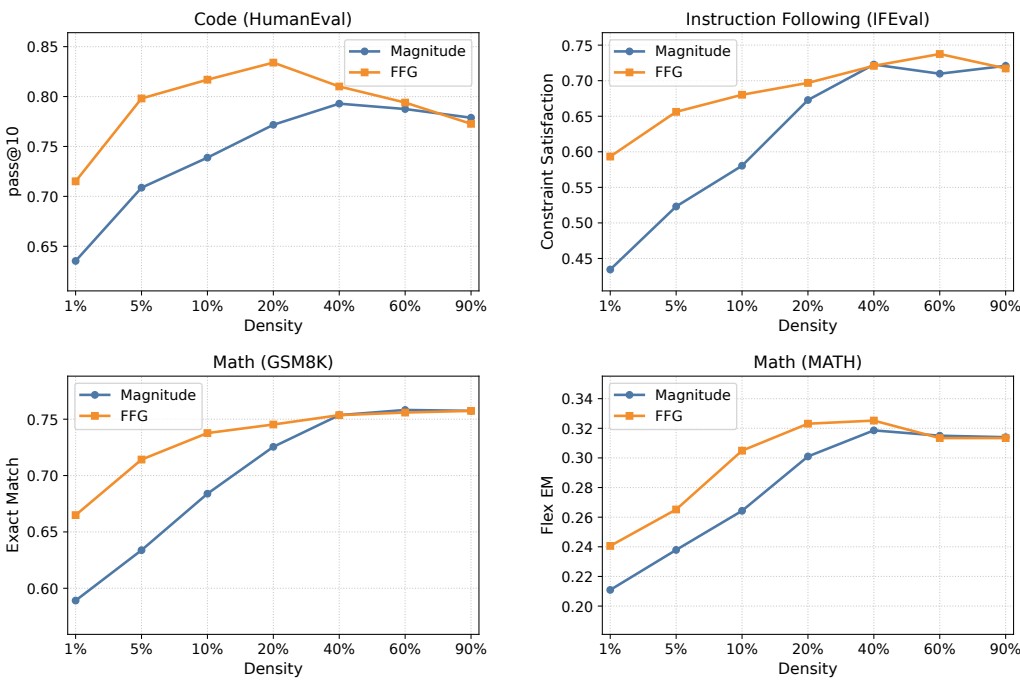

Figure 7: FFG vs. magnitude pruning across varying density ratios. FFG consistently outperforms, especially at lower densities (1-10%), highlighting its superior ability to identify salient parameters.

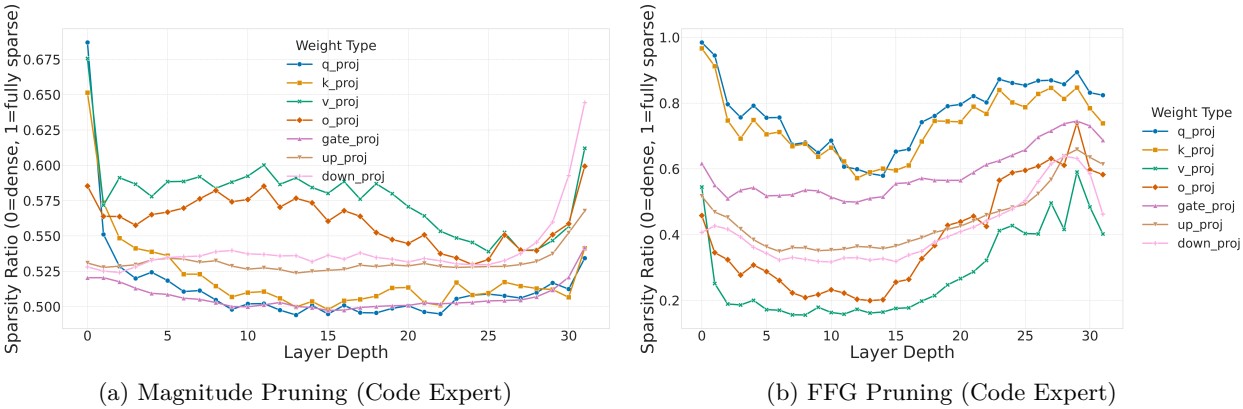

(a) Magnitude Pruning (Code Expert)          (b) FFG Pruning (Code Expert)

Figure 8: **Layer-wise density distribution at a global 40% task vector pruning density for the Code SFT expert.** FFG (right) exhibits strong, emergent role-aware pruning, aggressively sparsifying query/key weights while preserving value/output/FFN weights. Magnitude pruning (left) is far more uniform and less structured.

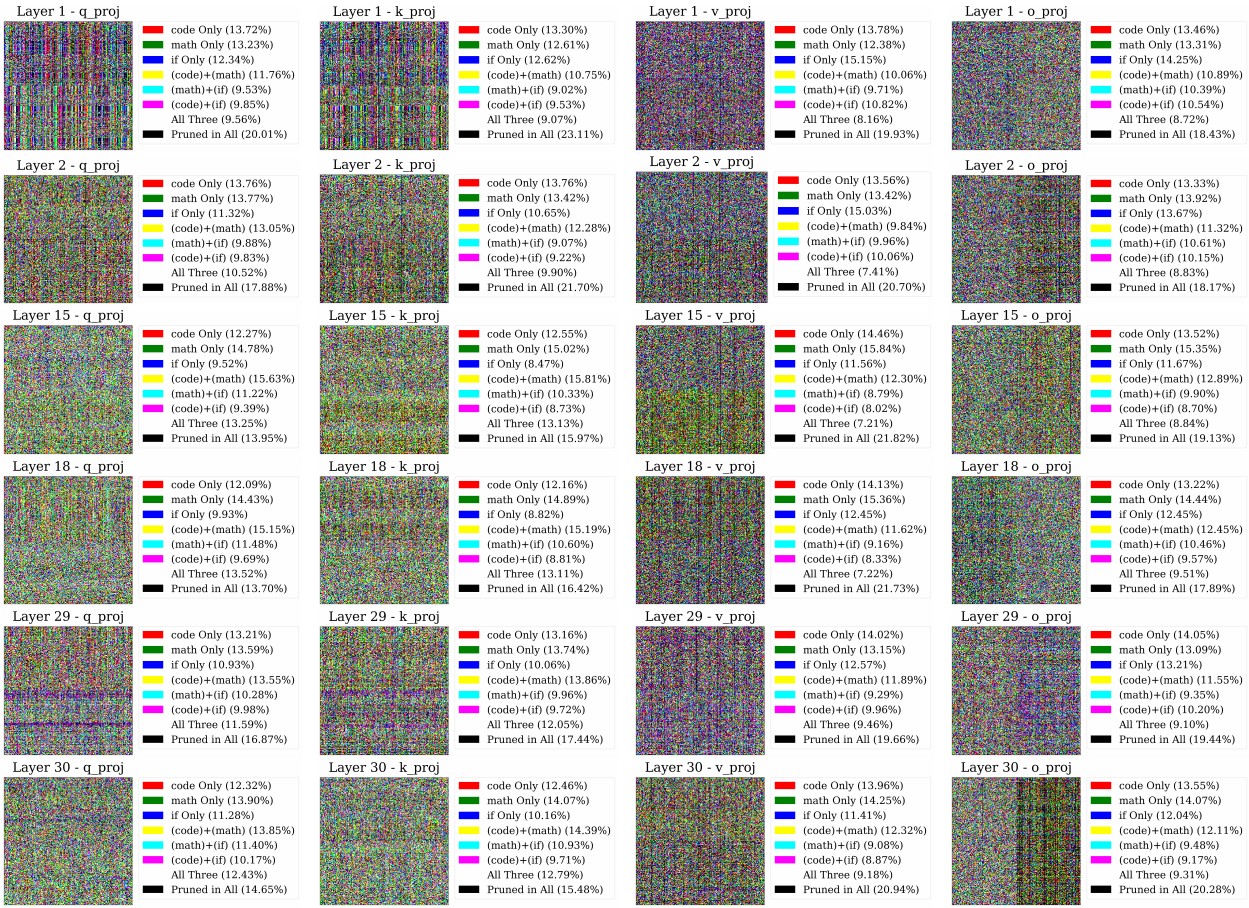

Figure 9: 3-way magnitude-based comparison for attention components across layers 1, 2, 15, 18, 29, and 30. Columns show $W_q$, $W_k$, $W_v$, and $W_o$.

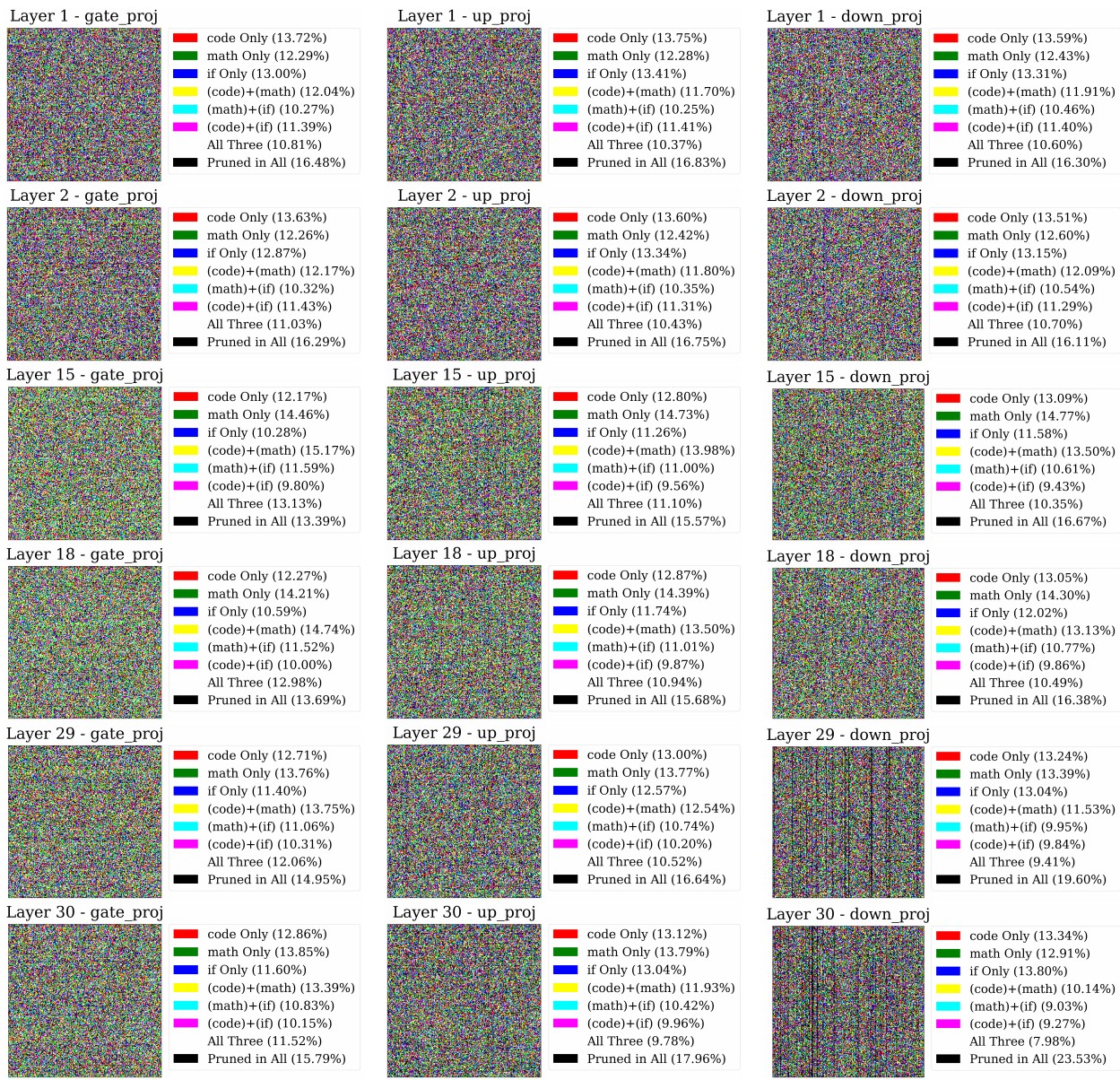

Figure 10: 3-way magnitude-based comparison for FFN components across layers 1, 2, 15, 18, 29, and 30. Columns show $W_{gate}$, $W_{up}$, and $W_{down}$.

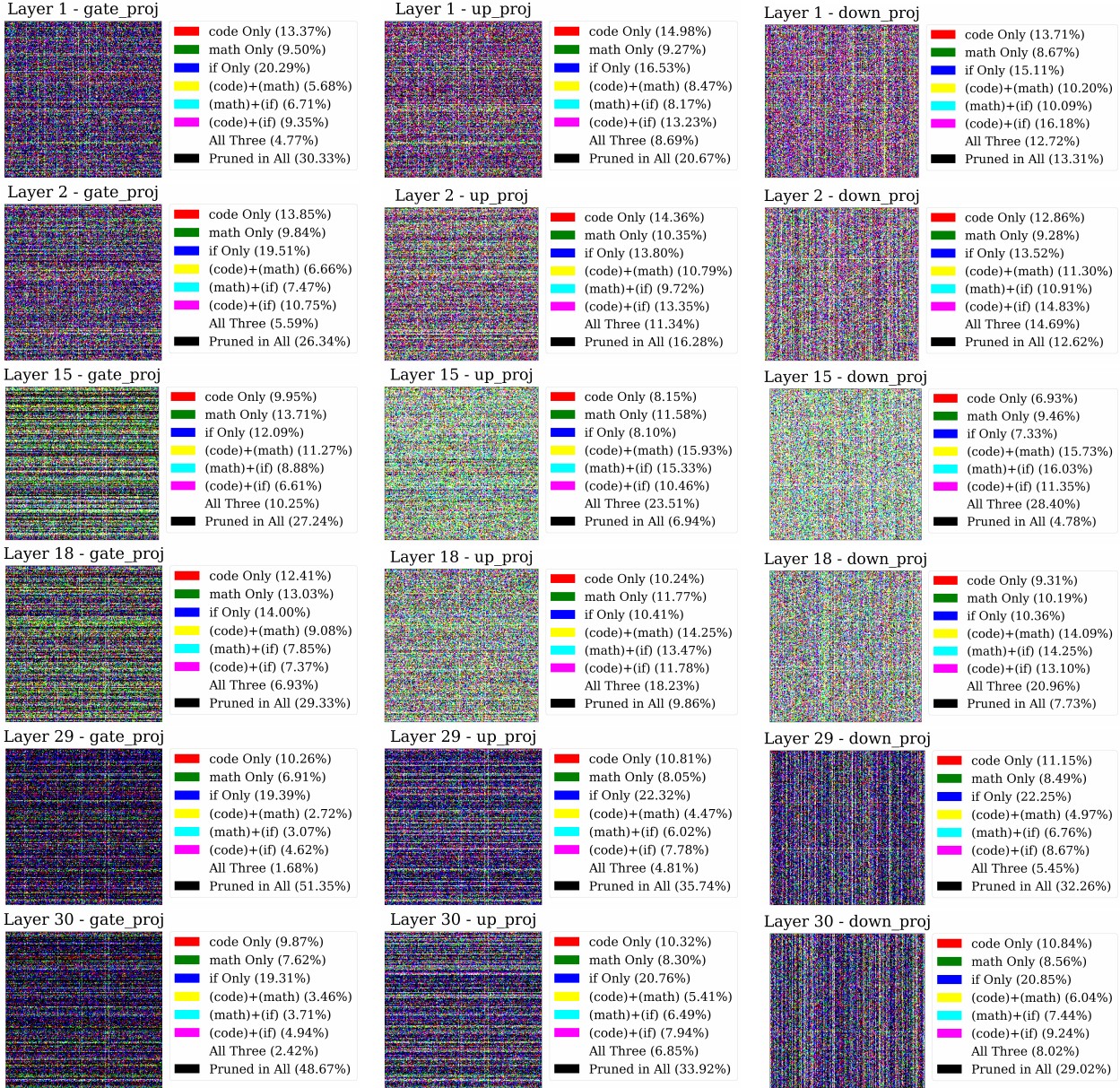

Figure 11: 3-way FFG comparison for FFN components across layers 1, 2, 15, 18, 29, and 30. Columns show $W_{gate}$, $W_{up}$, and $W_{down}$.

## E.1 Visualizing Structured Sparsity Masks

**What the histograms show.** For a weight matrix $W \in \mathbb{R}^{d_{out} \times d_{in}}$ with binary mask $M \in \{0, 1\}^{d_{out} \times d_{in}}$, we summarize mask structure via the row-wise and column-wise sparsities

$$\rho_i^{row} = 1 - \frac{1}{d_{in}} \sum_{j=1}^{d_{in}} M_{ij}, \qquad \rho_j^{col} = 1 - \frac{1}{d_{out}} \sum_{i=1}^{d_{out}} M_{ij},$$

i.e., the fraction of zeros in each row/column (sparsity $= 1 -$ density). Each panel in Figs. 12 and 13 plots the histogram of $\{\rho_i^{row}\}_{i=1}^{d_{out}}$ (top) and $\{\rho_j^{col}\}_{j=1}^{d_{in}}$ (bottom) for the self-attention $q$-projection of a single layer. A spike near 1.0 indicates rows/columns that are almost entirely pruned. All masks shown correspond to a global 40% density budget.

**Key finding: FFG induces structured channel sparsity at the network edges.** Under a single global density budget, FFG reallocates nonzeros across depth and weight types. In the $q$-projection, early (layers 0–1) and late (layers 29–30) blocks display pronounced structure: their histograms concentrate near $\rho \simeq 1.0$, revealing many rows/columns that are nearly all zeros (Fig. 12). A similar pattern is observed for the $k$-projection (not shown), indicating that FFG often eliminates entire input/output channels in these attention blocks rather than scattering zeros uniformly.

**Contrast with magnitude pruning.** For the same global budget, magnitude pruning yields weaker row/column structure: its histograms are centered around moderate sparsities with limited mass near 1.0 (Fig. 13). Thus, FFG is not merely more sparse; it is selectively sparse at the level of entire channels.

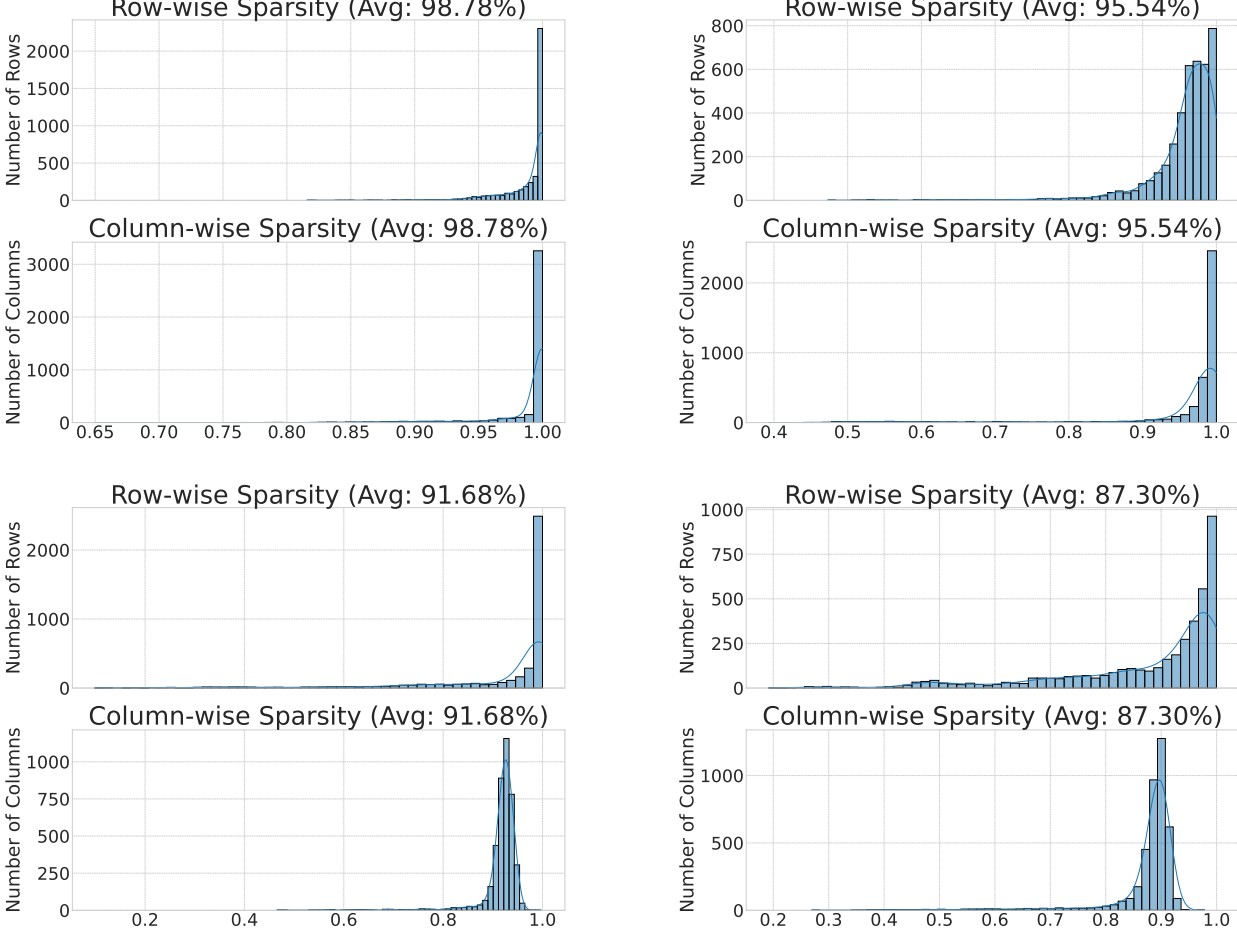

Figure 12: **FFG masks exhibit structured sparsity in the self-attention $q$-projection.** Row-wise (top) and column-wise (bottom) sparsity histograms for layers 0, 1, 29, and 30 (left-to-right, top-to-bottom). Note the concentration near $\rho \approx 1.0$, indicating that many rows/columns are almost entirely pruned.

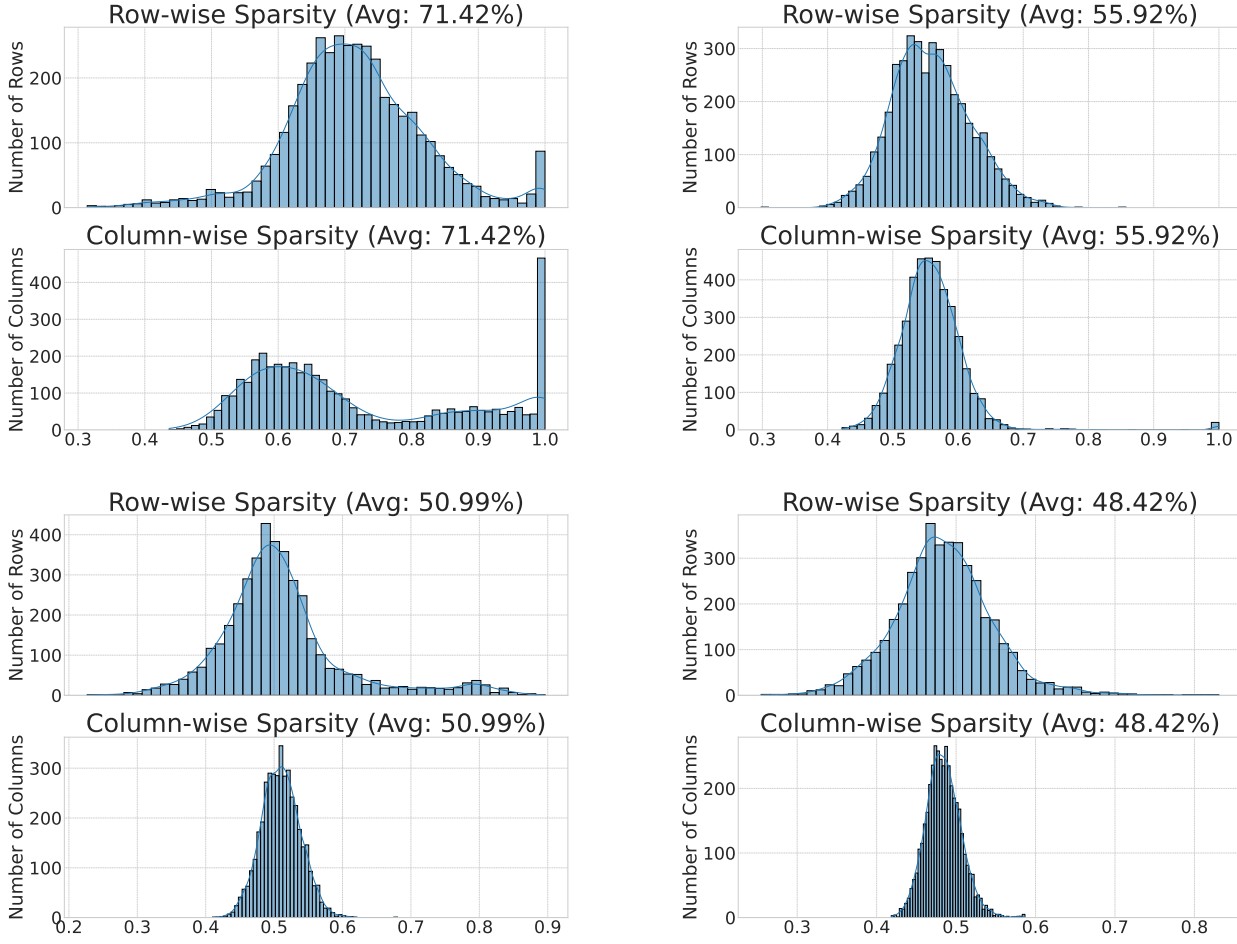

Figure 13: **Magnitude pruning produces weaker row/column structure.** Histograms for the same layers and weight type as Fig. 12 show mass centered at moderate sparsities and far less concentration near 1.0.

## F  Complementary Curvature, and Rank Analysis

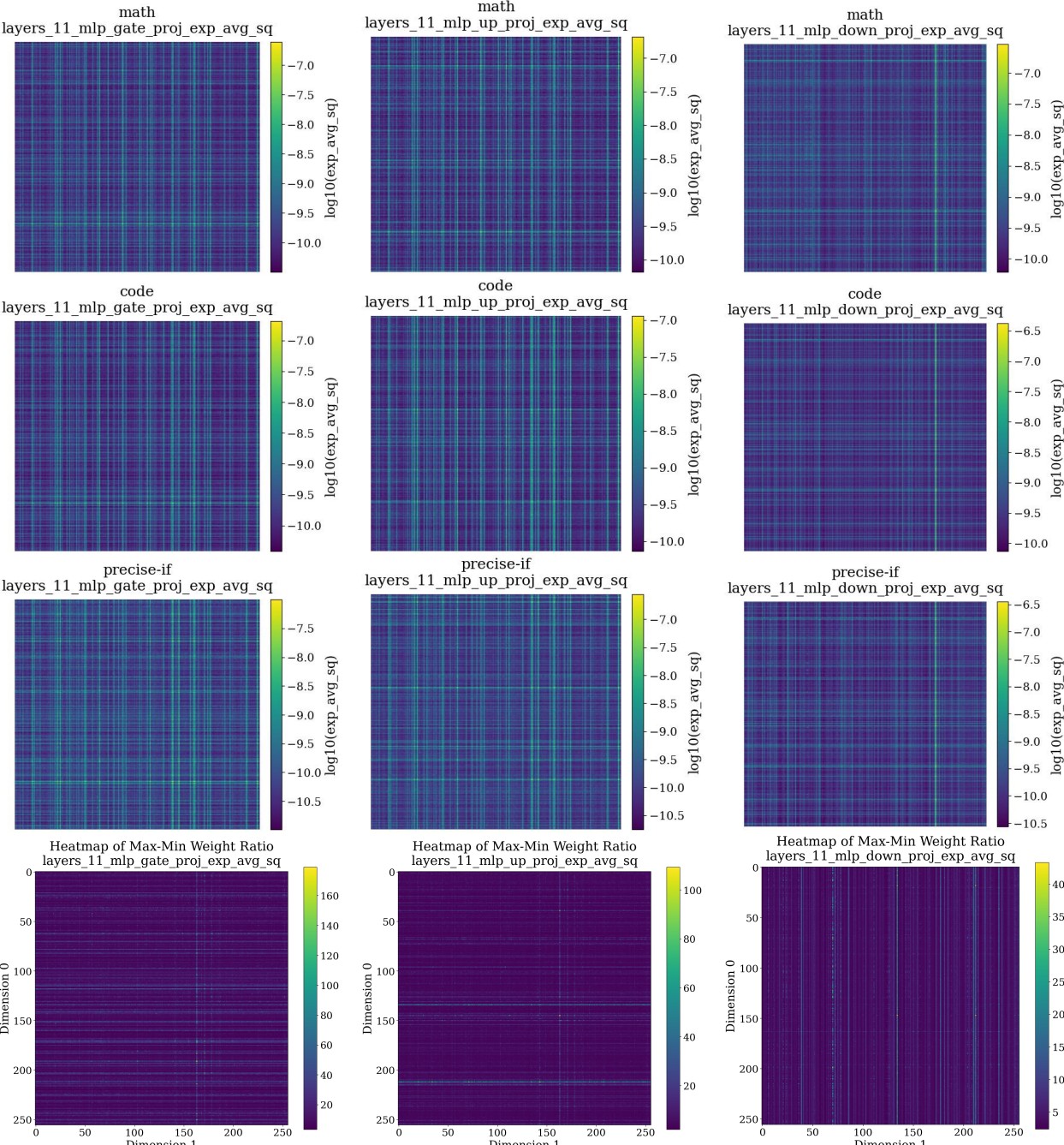

Figure 14: **Shared curvature geometry in FFN layers across specialist models.** Log-scaled heatmaps of the square root of the second-moment Adam statistics for layer 11 feed-forward network projection weights. Rows represent: Math specialist, Code specialist, Precise IF specialist, and Max-Min ratio across all models (top to bottom). Columns show $W_{gate}$, $W_{up}$, and $W_{down}$ (left to right). The structural similarity persists even in FFN layers, reinforcing our finding that shared geometry is a model-wide phenomenon. The bottom row quantifies the variance across models, with darker regions indicating higher consensus in curvature patterns.

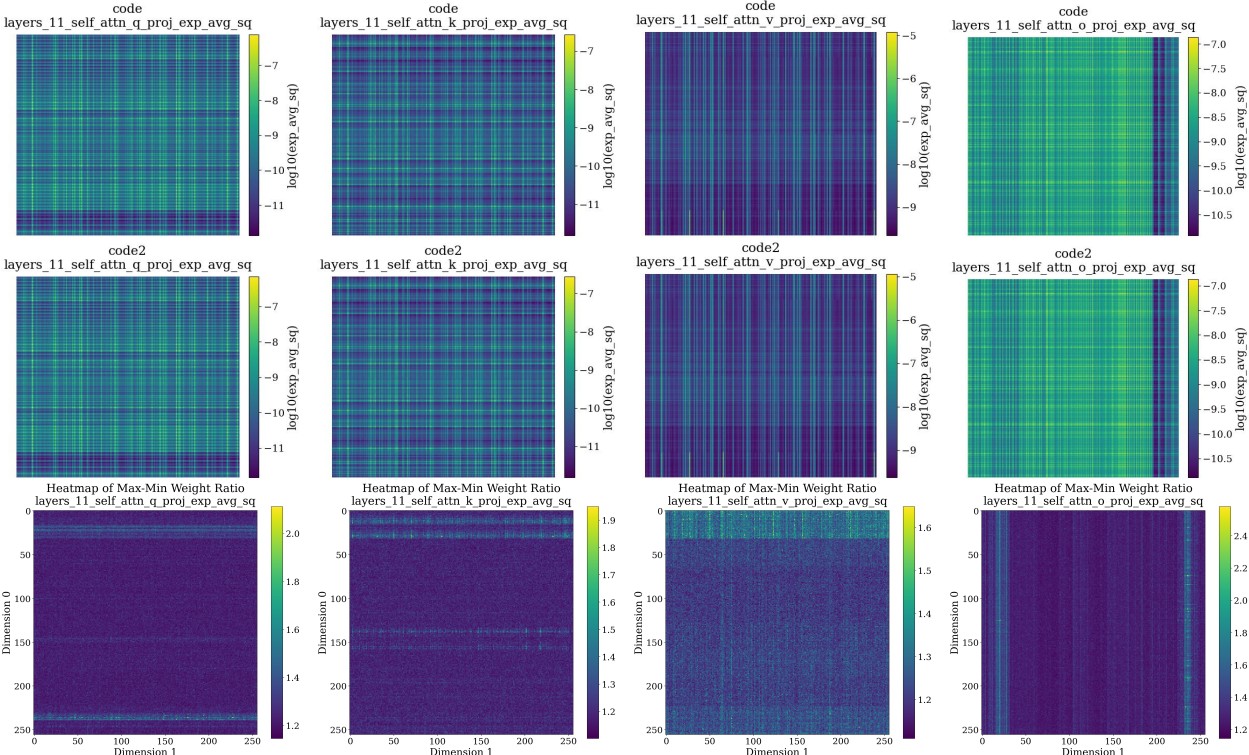

Figure 15: **Control Experiment: Shared curvature in attention layer 11 for two Code models.** The top row shows a Code model trained with a Cosine LR schedule, and the second row shows a Code model trained on the same data with a WSD schedule. The structural similarity is nearly perfect. The bottom row shows the max-min ratio is consistently close to 1 (dark color), indicating minimal geometric deviation.

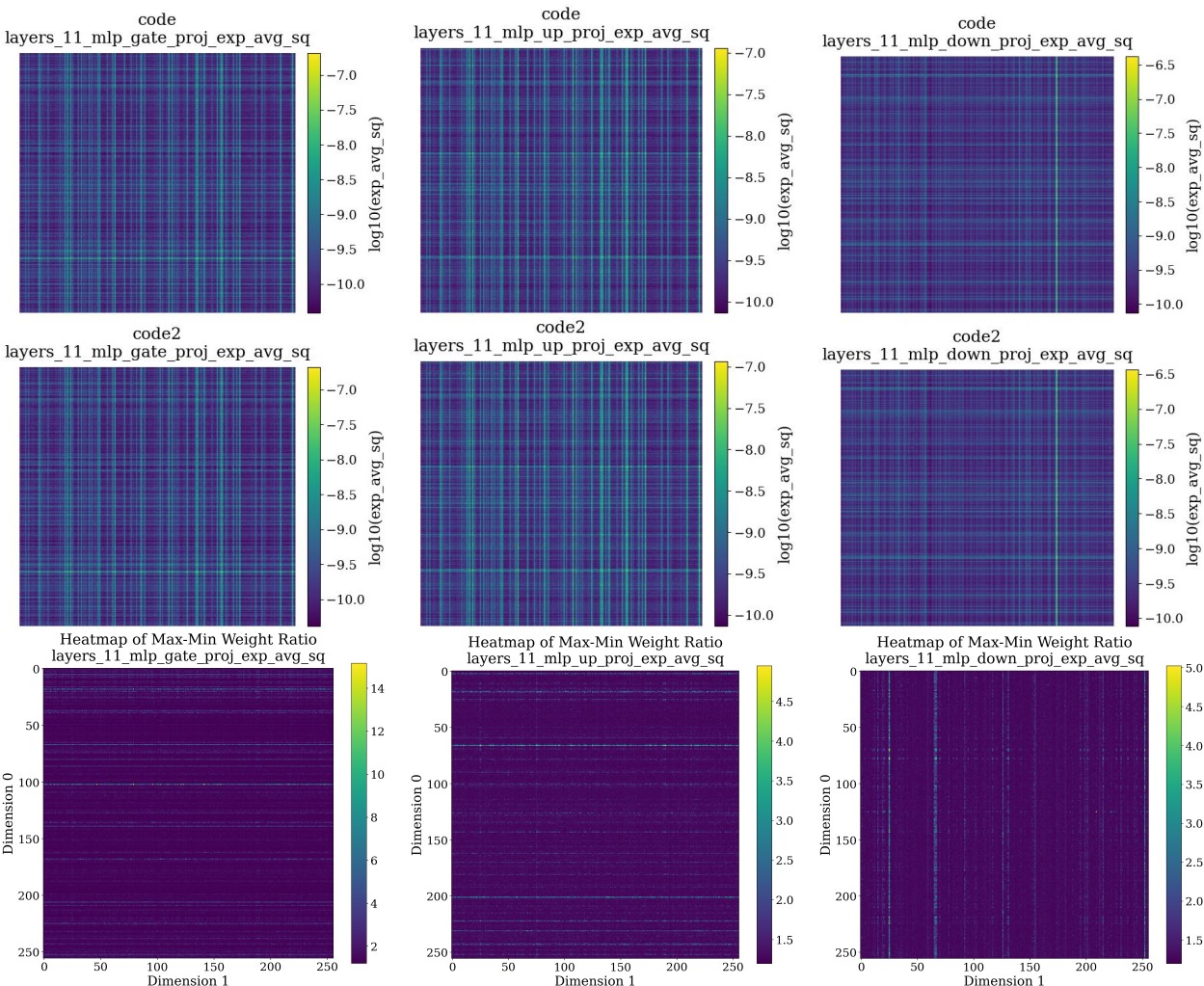

Figure 16: **Control Experiment: Shared curvature geometry in FFN layers for two Code models.** Log-scaled heatmaps of the square root of the second-moment Adam statistics for layer 11 feed-forward network projection weights. Rows represent: Code specialist (Cosine LR), Code specialist (WSD LR), and Max-Min ratio across the two models (top to bottom). The near-perfect structural similarity and low max-min ratio provide a strong control for our main hypothesis.

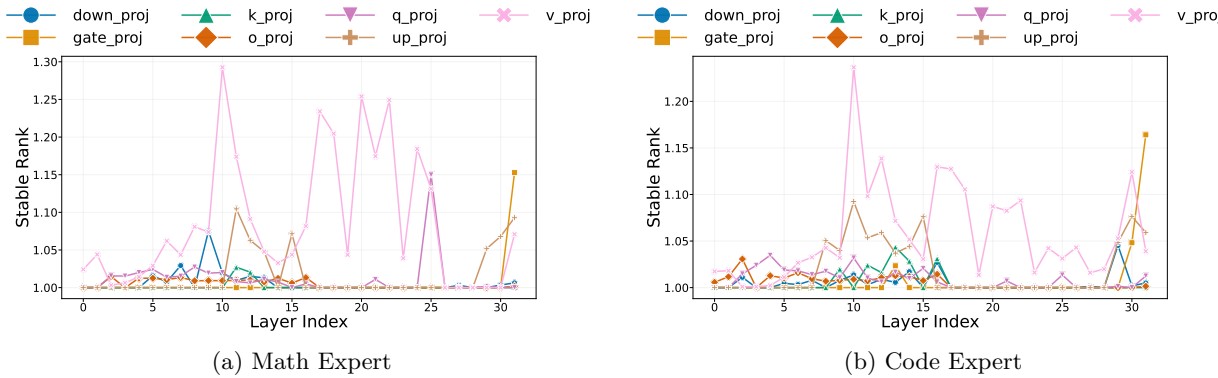

(a) Math Expert                                  (b) Code Expert

Figure 17: Stable rank analysis of the second-moment matrices ($\boldsymbol{v}_\tau$). The consistently low stable rank across all layers validates our use of AdaFactor for compression.

