# OpenReview forum: "Harnessing Optimization Dynamics for Curvature-Informed Model Merging"
_TMLR — Rejected by TMLR_

### Review · Reviewer_imwq · 2026-03-16

**Summary Of Contributions:**

This paper proposes a curvature-informed model merging method that improves the combination of multiple fine-tuned models by leveraging optimization dynamics and local curvature information of the loss landscape. Instead of naively averaging parameters, the method estimates curvature-aware importance weights that reflect how sensitive different parameters are during training. This allows the merging process to better preserve task-specific knowledge while reducing interference between models.

**Audience:**

Yes

**Audience Explanation:**

In general, this is a good add to the current model merging field. I see this as a *fully* 2nd order merging pipeline that aligns a few common merging techniques well: pruning to remove task conflict + 2nd order methods starting from Fisher merging. The reproduction from Tulu3 also makes the paper closer to practical usage given that many merging methods were evaluated on smaller scale tasks like GLUE.

**Broader Impact Concerns:**

NA.

**Claims And Evidence:**

No

**Claims Explanation:**

My major concern is that the analysis starting from 4.3 relies too much on Llama's architecture. For example, the exact region/layers (Figure 2,3) conclusion may change a lot across models (and probably tasks). Since the cost for the merging method + optimizer based fisher approximation is not very costly (especially the latter about 2nd order matrices), I recommend authors to do a more thorough analysis with existing model merging benchmarks with open-sourced weights. See requested changes below.

**Requested Changes:**

1. Check your \citet and \citep usage.
2. See "He, Yifei, et al. "MergeBench: A Benchmark for Merging Domain-Specialized LLMs." The Thirty-ninth Annual Conference on Neural Information Processing Systems Datasets and Benchmarks Track." and you can get access to their checkpoints on huggingface to repeat the same layer wise analysis in Sec. 4. I think this will support many of your claims better, as they support Llama2/8B and Gemma 3/9B on modern LLM tasks.
3. Sec 4.3.1, about the relationship with LoRA, I am curious about the effect of OTA and FFG on LoRA trained checkpoints since this is 1) a very popular model merging research area 2) you mentioned your method might be related with LoRA. I wonder if OTA-FFG will be still effective on LoRA compared to Fisher and other magnitude pruning methods, or the advantages just cancel out because LoRA and your proposed solution have the same function.

---

### Review · Reviewer_no2y · 2026-03-16

**Summary Of Contributions:**

This paper introduces OTA merging; a two-stage model merging method using the Adam optimizer's second-moment statistics as a proxy for diagonal curvature. It starts by scoring task-vector parameters with a "saliency" term that it is used to keep low-saliency parameters to their values in the base model. Then, it uses the curvature-aware weighting to merge the remaining parameters. Additionally, the paper proposes a compression method to reduce the memory cost of storing the second-moment tensors. In terms of results, the paper shows that the method outperforms other model merging methods after evaluating on subsets of the Tulu3 dataset.

**Audience:**

Yes

**Audience Explanation:**

The paper is clearly relevant to readers working on model merging, pruning/grafting, optimizer-state reuse, and post-training of LLMs. Even if I believe some claims need narrowing, the paper contains insights that would be valuable to a portion of the TMLR readership.

**Broader Impact Concerns:**

Nothing to report

**Claims And Evidence:**

No

**Claims Explanation:**

The paper provides convincing evidence that OTA is a promising merging method in the experimental setup studied. The method is well motivated, the reuse of optimizer second moments is practical and efficient, and the ablations suggest that the pruning step reduces interference between experts. However, I believe some claims are broader than what the evidence supports:

- The performance claim should be narrowed. A shown in Table 1, OTA gets the best average score, but is not the best across all capabilities and the gains over TIES are modest.
- The “new merging theory” claim appears overstated. The analysis provides interesting empirical evidence that optimizer second moments can be a useful diagonal curvature proxy, but the evidence is limited to diagonal approximations and one training pipeline. This supports an empirical hypothesis rather than a general theory.
- The curvature proxy claim could be framed more cautiously. The paper shows that Adam second moments work well as a proxy in this setting, but it does not validate this proxy against stronger curvature estimates.
- The sensitivity of the sparsity tuning protocol. Sparsity is selected per expert using the same capability benchmarks that are later used to evaluate the merged model. I might be wrong but, because the results are sensitive to sparsity,  this tuning protocol seems like it is indirectly optimizing on the evaluation tasks.

**Requested Changes:**

**Important**
- Narrow the strongest claims (e.g., “consistent outperformance,” “state-of-the-art,” and “new merging theory”) to reflect the experimental scope.
- Reframe the curvature analysis as an empirical finding unless stronger evidence is provided.

**Strengthening**
- Report uncertainty estimates (e.g., multiple runs or confidence intervals) for the main comparison results.
- Soften interpretability claims about task localization unless additional quantitative analysis is provided.
- If feasible, redo sparsity tuning using a held-out validation protocol separate from the final capability benchmarks, and report results under this protocol.

---

### Review · Reviewer_3TRi · 2026-03-17

**Summary Of Contributions:**

This paper introduces OTA (Optimization Trajectory Aware) Merging, a framework for merging SFT-specialized LLM checkpoints using Adam optimizer second-moment estimates as a diagonal curvature proxy. The framework consists of (1) Fast Fisher Grafting (FFG), a curvature-aware saliency pruning step that grafts non-salient parameters back to base weights; (2) curvature-aware aggregation via closed-form preconditioned weighted averaging; and (3) AdaFactor-style rank-1 compression of second moments (29.9 GB to 12.6 MB). Experiments merge five Llama 3.1-8B SFT specialists from the Tulu-3 pipeline, evaluated with OLMES toolkit.

**Key strengths:**

- **FFG is the standout contribution.** FFG-TA (FFG + linear averaging, 0.560) already surpasses most baselines, and full OTA (0.582) beats TIES (0.565). FFG exhibits emergent structured sparsity: it reverses magnitude pruning's density rankings, aggressively sparsifying Q/K projections while preserving V/O/FFN weights, with 85%+ row zeroing in early/late layer Q-projections. These patterns are genuinely surprising and connect naturally to LoRA-compatible low-rank structure.
- **The shared curvature hypothesis is well-supported.** Independently SFT'd models develop similar curvature structures. The control experiment (two Code models with different LR schedulers showing near-identical curvature) convincingly disentangles task-driven from artifact-driven differences, offering a new explanation for why linear averaging works.
- **Rank-1 compression is practical.** A 2,375x compression ratio with only 0.011 performance drop, justified by stable rank < 1.3 across all layers.
- **Task localization analysis** (Math dominating attention Q/K, Code/IF dominating FFN layers 22-31) provides useful interpretability insights.

**Key weaknesses:**

- **Curvature-aware aggregation hurts without FFG** (OTA w/o FFG = 0.536, worse than linear = 0.551). The paper overstates the aggregation's independent contribution.
- **Modest improvement over TIES** (1.7 points) with no confidence intervals reported.
- **Incremental novelty.** The Adam-as-Fisher proxy is established by Li et al. (2025); the grafting builds on Panigrahi et al. (2023); the aggregation is standard Fisher-weighted merging. The novelty lies in the combination and empirical analysis.
- **Single model family and scale** (Llama 3.1-8B only).
- **Per-expert sparsity tuning is critical** to performance, and a tuned-linear baseline is missing.

**Additional Comments:**

The paper's primary value lies not in the OTA aggregation formula (which is the weakest component) but in three complementary contributions: (1) FFG as a principled curvature-informed denoising tool with rich analysis of its emergent structured sparsity and task-localization properties; (2) the shared curvature hypothesis with supporting evidence; and (3) the practical rank-1 compression scheme. The analytical depth, particularly the controlled experiments and FFG vs. magnitude pruning analysis, exceeds what is typical in the merging literature.

The three critical requested changes (sensitivity analysis, reframing around FFG, per-expert-scaled linear baseline) are feasible within a minor revision cycle and would substantially strengthen the empirical claims without requiring new large-scale experiments.

**Audience:**

Yes

**Audience Explanation:**

The paper addresses a problem of growing practical importance (merging SFT-specialized LLMs) and appeals to multiple audiences:

- **Practitioners** gain a plug-and-play framework requiring only saved optimizer states, with rank-1 compression making storage overhead negligible.
- **Model merging researchers** gain the insight that interference mitigation via denoising (FFG) matters more than curvature-aware weighting, which could influence future method design.
- **Fine-tuning dynamics researchers** gain the shared curvature hypothesis, FFG-based task localization analysis, and the finding that FFG implicitly discovers LoRA-compatible low-rank structure.

The analytical depth (layer-wise density distributions, 3-way mask visualizations, curvature heatmaps, control experiments) provides substantially more insight than a typical methods paper.

**Broader Impact Concerns:**

None. This work addresses model merging for SFT-specialized LLMs, a standard technique in open-source model development. The method does not introduce new capabilities beyond what the constituent models already enable. Open-sourcing of code, checkpoints, and evaluation scripts supports reproducibility.

**Claims And Evidence:**

Yes

**Claims Explanation:**

1. **"OTA achieves state-of-the-art merging performance"**: Supported. Table 1 shows OTA (0.582) outperforms all baselines including TIES (0.565), with gains across Math (+0.029), Code (+0.035), and Instruction (+0.035). The comparison is fair (both use per-expert sparsity tuning with the same search budget). The consistent advantage across 4 of 5 categories and the clean ablation ordering (OTA w/o FFG < Linear < FFG-TA < OTA) provide indirect evidence of robustness, though the 1.7-point margin would benefit from sensitivity analysis.

2. **"FFG is the critical component"**: Strongly supported. FFG-TA (0.560) substantially outperforms OTA w/o FFG (0.536), and OTA w/o FFG performs *worse* than linear averaging (0.551). This is the paper's most convincing empirical finding.

3. **"Shared curvature geometry across SFT models"**: Well-supported at the tested scale. Element-wise max-min ratios stay below 5 despite within-model curvature spanning 4 orders of magnitude. The Code-Cosine vs. Code-WSD control experiment is particularly convincing, though this remains demonstrated only for Llama 3.1-8B.

4. **"Rank-1 compression is viable"**: Strongly supported. Stable rank < 1.3 across all layers directly justifies the approximation; the performance drop is only 0.011.

5. **"Curvature-aware aggregation improves merging"**: Only partially supported. The aggregation helps when combined with FFG (+0.022 over FFG-TA) but actively hurts when applied alone (-0.015 vs. linear). The paper does not adequately acknowledge this asymmetry.

**Requested Changes:**

1. **(Critical)** **Report sensitivity analysis for Table 1.** At minimum, report performance variation when perturbing the per-expert density ratios by +/-5% around the selected values for OTA and TIES. If feasible, reporting results across 2-3 independently trained SFT checkpoint sets would further strengthen the statistical argument.

2. **(Critical)** **Reframe the narrative to center FFG as the primary contribution.** The ablation evidence clearly shows FFG is the main performance driver. Sections 1 and 3 should present FFG as the core innovation and aggregation as a complementary component. Explicitly acknowledge that OTA w/o FFG underperforms linear averaging.

3. **(Critical)** **Add a "linear merging with per-expert scaling coefficients" baseline.** Tuning $\alpha_\tau$ in $w_0 + \sum \alpha_\tau \Delta w_\tau$ over the same search budget would isolate how much of OTA's advantage comes from per-expert tuning vs. FFG's parameter selection.

4. **(Non-critical)** **Scope generalizability claims.** Qualify the abstract and conclusion to reflect that the method has been validated on Llama 3.1-8B only. Acknowledge that multi-architecture and multi-scale validation is needed.

5. **(Non-critical)** **Discuss the aggregation-alone failure mode.** Explain why curvature-aware aggregation hurts on unpruned task vectors (e.g., the preconditioner amplifies noisy, low-saliency parameters without FFG's filtering).

6. **(Non-critical)** **Address missing baselines.** EMR-merging (Huang et al., 2024, NeurIPS Spotlight) and MaTS (Tam et al., 2024) are cited but not benchmarked. Either include them or explain why they were excluded.

---

### Review · Reviewer_BtoF · 2026-03-17

**Summary Of Contributions:**

This paper proposes OTA-Merging, a framework for merging multiple SFT expert models using Adam's second-moment statistics (`exp_avg_sq`) as a diagonal curvature proxy, specifically as a training-free approximation to the diagonal Fisher Information Matrix. The framework has three components: (1) Fast Fisher Grafting (FFG), which scores each parameter's task-vector change by a curvature-weighted saliency ($s_{\tau,i} = v_{\tau,i} \cdot (\Delta w_{\tau,i})^2$) and reverts low-saliency updates back to the base model weights, (2) curvature-aware aggregation, which merges the denoised task vectors via a preconditioned weighted average, and (3) AdaFactor-style rank-1 compression of the second-moment tensors.

Beyond the merging method, the paper includes two further empirical contributions: a task localization analysis via FFG masks that uncovers structured, role-aware sparsity patterns (aggressive Q/K pruning vs. dense V/O/FFN in middle layers, separation of general-purpose and task-specific attention heads), and a Shared Curvature Hypothesis supported by heatmap comparisons showing that independently fine-tuned experts lies in basins with similar diagonal curvature geometry.

**Key strengths:**
- Reuse of existing optimizer state at no extra cost
- FFG's regularization effect: a sparse task vector can outperform the dense one
- Task localization findings
- Empirical support for the shared curvature hypothesis
- Controlled, reproducible setup based on Tülu-3

**Key weaknesses:**
- Gap between the theoretical claim ($v_T \approx \text{diag}(\bar{F}(w^\star))$) and the actual EMA dynamics during training
- OTA without FFG underperforms simple linear averaging, with no satisfying explanation
- Missing ablation of FFG paired with a directly computed diagonal FIM

**Audience:**

Yes

**Audience Explanation:**

The paper contributes to three areas the ML community currently is interested in.

**Model merging.** Combining specialized models without re-training has clear practical value. Results at a realistic scale (LLaMA 3.1-8B, Tülu-3 recipe) are useful to practitioners working on post-training.

**Optimizer statistics as a curvature proxy.** The observation that `exp_avg_sq` carries geometric information useful outside of optimization opens directions in model merging. Li et al. (2025) made a similar point. This paper goes further by applying it to the SFT merging problem.

**SFT interpretability via task localization.** FFG masks reveal structured, role-aware sparsity: Q/K layers are low-rank, V/O/FFN in the middle carry most task-specific information, and late-layers split into general and task-specific heads. This gives a concrete picture of what SFT actually modifies in a pretrained model.

The Shared Curvature Hypothesis also addresses a question that has not been resolved yet in the merging literature: why does linear weight averaging work as well as it does? The paper partially gives a answer to this.

**Broader Impact Concerns:**

None specific to this work.

**Claims And Evidence:**

Yes

**Claims Explanation:**

**My answer: Yes, but with minor reservations.**

- The main empirical claims are quite compelling. OTA outperforms baselines (avg. 0.582 vs. TIES 0.565 vs. Linear 0.551) on a diverse, controlled benchmark suite. The FFG regularization result is notable: at 20% density, FFG achieves HumanEval 0.834, above the dense SFT model's 0.788, a result that holds independently of merging. The task localization findings are supported by layer-wise and role-wise visualizations across multiple models, covering U-shaped sparsity, Q/K vs. V/O density inversion, and the two-region behvaior in late-layer value/output projections. The Shared Curvature Hypothesis reliies on a concrete quantitative comparison. The cosine vs. WSD experiment separates the effect of task data from optimizer schedule.

- Two gaps are also observed. The first concerns the theoretical arguments connecting $v_T$ to the diagonal FIM at $w^\star$. Lemma 2 characterizes the second moment of mini-batch gradients computed at a fixed $w^\star$, but Adam's $v_T$ is an EMA of squared gradients accumulated *across the entire trajectory*, where each $g_k$ is computed at a different $w_k$. These are not the same quantity, and the paper never addresses this distinction. Under Adam's default $\beta_2 = 0.999$, recent gradients dominate. Mini-batch gradient norms do not directly indicate non-convergence of the full-dataset loss, but this at least raises doubt about how closely the final iterates approximate $w^\star$.

- There is also an another issue on theoretical arguments. Since Adam's gradients are always computed with observed labels, $v_T$ approximates $\text{diag}(\bar{F})$, not $\text{diag}(F)$. $\bar{F}$ is a reasonable proxy for parameter importance, and this does not weaken the method itself. However, the paper's use of "curvature proxy" throughout implies a connection to the true loss landscape curvature that is only formally warranted under the perfect calibration assumption. The theoretical arguments should be more rigorous about this.

- The natural experiment to address the this issue is to compute the diagonal empirical FIM directly at $w^\star$ via one full forward pass over the training set (as FIM), accumulating $\left(\partial \log p(y_i|x_i;w^\star)/\partial w_j\right)^2$ for each parameter. This is exactly what Lemma 2 characterizes and is distinct from $v_T$. If performance is similar to OTA with $v_T$, the proxy is justified. If substantially better, the sufficiency claim would need to be qualified.

**Requested Changes:**

### Critical

**[C1] The Adam EMA is not the diagonal empirical FIM at $w^\star$, and the paper does not address this**

The paper's theoretical argument proceeds as follows. Lemma 1 shows Hessian $\approx$ empirical FIM at $w^\star$ under Late NTK Locality and Perfect Calibration. Lemma 2 shows that the expectation of mini-batch gradient outer products at $w^\star$ equals $\frac{1}{|B|}\bar{F}(w^\star)$. The paper then concludes that Adam's EMA $v_T \approx$ diagonal FIM at $w^\star$. The first two steps are proven, but the move to the third is never justified.

Adam's $v_T$ is defined as:

$$v_T = (1 - \beta_2) \sum_{k=0}^{T-1} \beta_2^{T-1-k}\, g_k^2$$

where $g_k = \nabla \mathcal{L}_{B_k}(w_k)$ is the gradient at $w_k$, which changes at every step. Lemma 2 instead characterizes gradients at a fixed $w^\star$. These are different quantities. Even with Adam's default $\beta_2 = 0.999$, the parameters at those final steps may differ meaningfully from $w^\star$. Figure 5 shows that mini-batch gradient norms still remains far from zero (ex. `knowledge_recall`) throughout training, which at least raises doubt about how closely the final iterates approximate $w^\star$.

A secondary issue is about the theoretical result. The empirical FIM $\bar{F}$ uses observed training labels, while the true FIM $F$ integrates over the model's own predictive distribution (the authors already note in Appendix). They coincide only under perfect calibration. Since Adam's gradients are always computed with observed labels, $v_T$ approximates $\text{diag}(\bar{F})$, not $\text{diag}(F)$. This does not invalidate the method, but the paper's use of "curvature proxy" throughout implies a connection to the true loss landscape curvature that is only formally guaranteed under the perfect calibration assumption, which is never verified. The theoretical support (or empirical support) should be more explicit about this.

The most direct way to address the primary issue is to compute the diagonal empirical FIM at the converged checkpoint directly: run one full forward pass over the training set at $w^\star$ and accumulate $\left(\partial \log p(y_i|x_i;w^\star)/\partial w_j\right)^2$ for each parameter. This is exactly what Lemma 2 characterizes and is distinct from $v_T$. Merging is a one-time operation, so I can understand though this cost would be quite heavy. If merging performance is similar to OTA with $v_T$, the proxy is justified. If substantially better, the sufficiency claim would need to be qualified.

**[C2] Ablation: FFG + directly computed diagonal FIM** *(continued from C1)*

The proposed experiment in C1 also motivates a direct ablation: compare OTA using $v_T$ against OTA using the diagonal empirical FIM computed via a full forward pass at $w^\star$. Table 1 provides indirect evidence here — Fisher Merging (no FFG, computed FIM, 0.541 to my understanding) vs. OTA w/o FFG (no FFG, Adam proxy, 0.536) shows a gap of only 0.005, suggesting that two proxies are nearly equivalent for the aggregation step. However, this does not answer whether the proxy choice matters for FFG saliency scoring, i.e., whether the mask $m_\tau$ changes meaningfully when $v_T$ is replaced by the directly computed FIM. This ablation would separate the contribution of FFG from the contribution of the curvature source.

**[C3] Why does OTA without FFG underperform linear averaging?**

Table 1 shows OTA (w/o FFG) at 0.536, below Linear at 0.551. The paper attributes this to task interference, but that does not explain why curvature reweighting is *worse* than uniform weighting. One plausible mechanism: when task vectors conflict in direction, high-curvature parameters receive amplified weights under $P^\star_\tau$, concentrating interference at the coordinates where tasks disagree most.

This result also creates a little inconsistency with the Shared Curvature Hypothesis. If $P^\star_\tau \approx P^\star$ for all $\tau$, as the shared curvature evidence suggests, the curvature-weighted aggregation reduces to approximately linear averaging. However, it performs below linear averaging without FFG. The paper does not explain this. The authors should either (a) provide a more thorough analysis of why the aggregation step degrades performance without FFG, or (b) discuss the conditions under which shared curvature does and does not imply parity with linear averaging.

**[C4] Missing geometric characterization of expert distances**

Both the FFG saliency step and the curvature-aware aggregation step depends on local second-order approximations around each expert's optimum $w^\star_\tau$. The paper never measures the scale of the perturbations involved, making it impossible to assess whether these approximations hold.

Two measurements would address this. First, $\Vert \Delta w_\tau \Vert_2 = \Vert w^\star_\tau - w_0\Vert_2$ per expert characterizes the scale of the task-vector perturbation that the OBD saliency approximation is applied to. The second-order Taylor expansion is valid only when $|\Delta w_{\tau,i}|$ is small relative to the curvature radius $1/\sqrt{v_{\tau,i}}$. If $\Vert \Delta w_\tau\Vert_2$ is large, the quadratic approximation may not hold.

Second, $\Vert w^\star_{\tau_1} - w^\star_{\tau_2} \Vert_2$ for each pair of experts characterizes whether the curvature at $w^\star_\tau$ is representative of the region where $w_{\text{merged}}$ lies. For example, if the ratio:

$$\frac{\Vert w^\star_{\text{math}} - w^\star_{\text{code}}\Vert_2}{\Vert \Delta w_{\text{math}}\Vert_2}$$

is large, the experts are far more separated from each other than from $w_0$, and the curvature at each $w^\star_\tau$ may not be a reliable guide to the geometry at the merged point. Both measurements require only a single norm computation on saved checkpoints.

To my understanding, the Shared Curvature Hypothesis is interesting hypothesis, but it does not fully resolve this concern in the current shape. One could argue that uniform curvature across parameter space makes the local approximation globally valid. But this loops back to C2: uniform curvature implies curvature-aware aggregation $\approx$ linear averaging, yet OTA (w/o FFG) underperforms linear averaging. Two findings are in little contradiction, and measuring expert distances would help clarify which picture is actually correct.

### Optional (would strengthen the work)

**[O1] Sparsity tuning overhead**

The paper acknowledges in Section 5 that sparsity ratios must be tuned per expert, using 5 density ratios $\in \lbrace 0.05, 0.1, 0.2, 0.4, 0.6 \rbrace$. This means adding a new capability requires 5 additional evaluations per merging method. A brief discussion of whether a proxy metric (e.g., task-vector norm or gradient norm at convergence) could predict the appropriate sparsity without evaluation would improve practical usability.

**[O2] Forgetting**

OTA scores 0.315 on PopQA vs. 0.318 for Linear and TIES. The difference (0.003) is small and likely within noise, but the pattern is consistent across all OTA variants: curvature-aware merging shows no meaningful improvement on knowledge retention. A brief note on whether this is expected.

**[O3] Theoretical account of the Shared Curvature Hypothesis**

The empirical evidence is quite solid I think, but there is no explanation for why different SFT tasks produce similar curvature structure. Even a short conjecture, such as that the pretrained model's geometry dominates and SFT causes only localized shifts, would give readers insights.

---

### Decision · Action_Editor_Wz2S · 2026-05-04

**Recommendation:** Reject

**Additional Comments:**

The authors did not submit a revision or substantively participate in the discussion, so the reviewers’ concerns remain unresolved. That said, I would be happy to handle a future resubmission, possibly with the same reviewers, if the authors choose to revise and resubmit.

**Audience:**

Yes

**Audience Explanation:**

Yes. The topic is relevant to TMLR readers interested in model merging and LLM post-training, and the findings seem useful despite the paper’s evidential limitations.

**Claims And Evidence:**

No

**Claims Explanation:**

The paper shows promising results, but, as the reviewers pointed out, several key claims remain insufficiently supported. The curvature-proxy argument is not convincingly validated, the gains over baselines are modest and lack sensitivity/uncertainty analysis, and the evidence is limited to one model family and scale.

**Resubmission Of Major Revision:**

The authors may consider submitting a major revision at a later time.